# Rif1 *S*-acylation mediates DNA double-strand break repair at the inner nuclear membrane

Gabriele A. Fontana [1,4], Daniel Hess [1,7], Julia K. Reinert [1,2,7], Stefano Mattarocci[3,5,7], Benoît Falquet [1,2], Dominique Klein[1], David Shore [3], Nicolas H. Thomä [1] & Ulrich Rass [1,6]

Rif1 is involved in telomere homeostasis, DNA replication timing, and DNA double-strand break (DSB) repair pathway choice from yeast to human. The molecular mechanisms that enable Rif1 to fulfill its diverse roles remain to be determined. Here, we demonstrate that Rif1 is *S*-acylated within its conserved N-terminal domain at cysteine residues C466 and C473 by the DHHC family palmitoyl acyltransferase Pfa4. Rif1 *S*-acylation facilitates the accumulation of Rif1 at DSBs, the attenuation of DNA end-resection, and DSB repair by non-homologous end-joining (NHEJ). These findings identify *S*-acylation as a posttranslational modification regulating DNA repair. *S*-acylated Rif1 mounts a localized DNA-damage response proximal to the inner nuclear membrane, revealing a mechanism of compartmentalized DSB repair pathway choice by sequestration of a fatty acylated repair factor at the inner nuclear membrane.

[1] Friedrich Miescher Institute for Biomedical Research, Maulbeerstrasse 66, CH-4058 Basel, Switzerland. [2] Faculty of Natural Sciences, University of Basel, Petersplatz 10, CH-4003 Basel, Switzerland. [3] Department of Molecular Biology and Institute of Genetics and Genomics in Geneva (iGE3), University of Geneva, 30 Quai Ernest-Ansermet, CH-1211 Geneva, Switzerland. [4] Present address: Department of Health Sciences and Technology, ETH Zürich, Schmelzbergstrasse 9, CH-8092 Zürich, Switzerland. [5] Present address: Institut de Biologie Physico-Chimique, UMR8226, Laboratoire de Biologie Moléculaire et Cellulaire des Eucaryotes, Sorbonne Université, CNRS/UPMC, 13 rue Pierre et Marie Curie, 75005 Paris, France. [6] Present address: Genome Damage and Stability Centre, School of Life Sciences, University of Sussex, Falmer, Brighton BN1 9RQ, UK. [7] These authors contributed equally: Daniel Hess, Julia K. Reinert, Stefano Mattarocci. Correspondence and requests for materials should be addressed to U.R. (email: U.W.Rass@sussex.ac.uk)

Rif (Rap1-interacting factor 1) supports diverse biological functions. First identified in *Saccharomyces cerevisiae* as a telomere-binding protein, Rif1 regulates telomere length by counteracting telomerase recruitment and attenuates DNA end-resection at dysfunctional telomeres[1]. These activities depend on telomere recruitment by Rap1, mediated by two Rif1 C-terminal regions, the RBM (Rap1-binding motif) and CTD (C-terminal domain), and a conserved N-terminal domain with intrinsic DNA-binding activity known as the HOOK domain[2–4]. Cooperative binding of Rif1 to DNA ends produces a protective protein sheath, which excludes DNA end-processing factors including telomerase and the DNA end-resection machinery[1,3].

Rif1 also serves as a regulator of DNA replication origins, a role that is conserved from yeast to human[5–9]. RVxF/SILK motifs mediate interactions between Rif1 orthologs and protein phosphatase 1 (PP1)[1]. Rif1 targets PP1 to replication origins, leading to removal of activating phosphorylations on components of the replication machinery, locally attenuating origin firing and modulating replication timing globally[10–15].

In recent years, Rif1 has emerged as a critical regulator of DSB repair pathway choice[1,16–28]. In mammalian cells, the RIF1–53BP1 axis antagonizes BRCA1-CtIP-mediated 5′-DNA end-resection, a process that exposes 3′-DNA overhangs for homologous recombination (HR)-dependent DSB repair. Thus, RIF1 helps stabilize DSB ends, promoting repair by re-ligation along the NHEJ repair pathway. The conserved N-terminal part of Rif1 plays a crucial role in attenuating DNA end-resection from yeast to human[1,3,18], and in yeast, the protective encapsulation of DNA by the HOOK domain provides a mechanistic rationale for this Rif1 activity[1].

Posttranslational modifications have been implicated in the regulation of Rif1 functions. For example, phosphorylation of Rif1 close to the RVxF/SILK PP1-binding sites disrupts Rif1-PP1 interactions, leading to the activation of Rif1-repressed replication origins[10–12]. RIF1 ubiquitination and SUMOylation are required for the timely dissociation of 53BP1-RIF1 complexes from DNA damage, access of BRCA1-CtIP, and DSB repair by HR[29,30]. Rif1 S-acylation has been detected in budding yeast[31]. Pfa4 (protein fatty acyltransferase 4), a member of the evolutionary conserved DHHC (aspartate–histidine–histidine–cysteine) family of protein acyltransferases[32] promotes Rif1 S-acylation at as-yet unmapped sites[33]. S-acylation, predominantly by covalent attachment of 16-carbon fatty acid palmitate moieties to cysteine residues by thioester linkage, increases the interactions of proteins with cellular membranes by providing membrane anchors. Consequently, S-acylation can affect protein localization, stability, and activity[34,35]. Loss of Pfa4 was shown to lead to decreased Rif1-nuclear membrane interactions and an increase in nuclear-luminal Rif1 at the expense of Rif1 at the nuclear periphery[33]. The nuclear periphery contains heterochromatin and yeast telomeres[36], and the redistribution of Rif1 in absence of Pfa4 correlated with transcriptional changes of heterochromatic and telomeric genes[33]. On the other hand, Rif1 telomere occupancy and the ability of Rif1 to maintain telomere length were unaffected upon deletion of PFA4 (ref. [33]). Since Pfa4 is also dispensable for the ability of Rif1 to regulate origin firing[8], the physiological roles of Rif1 S-acylation remain to be fully elucidated[33,37].

Here, we show that Pfa4-dependent S-acylation of Rif1 promotes the attenuation of DNA end-resection and DSB repair by NHEJ, implicating protein S-palmitoylation in DNA repair. Mapping and mutating Rif1 S-acylation acceptor sites, we find that fatty acylation enables a localized DNA-damage response at the nuclear membrane and reveal a mechanism of compartmentalized DSB repair pathway choice.

## Results

**The palmitoyl acyltransferase Pfa4 promotes NHEJ.** Rif1 fulfills an evolutionarily conserved function in DSB repair pathway choice, stabilizing DNA ends and promoting DSB repair by simple re-ligation along the NHEJ pathway[3,16–20]. We hypothesized that if the reported Pfa4-dependent S-acylation of S. cerevisiae Rif1 (ref. [33]) is important for Rif1's role in NHEJ, deletion of PFA4, like a deletion of RIF1, should decrease the efficiency of NHEJ. To test this, we used a reporter strain containing an inducible DSB at the MAT locus that can only be repaired by NHEJ[38] (see Fig. 1a for details). As expected, cell viability upon DSB induction was fully dependent on core NHEJ factor Ku70 (Fig. 1b). Consistent with previous results[3], deleting RIF1 led to a marked decrease in cell viability by ~40% after 2 h of transient DSB induction, reflecting compromised NHEJ in absence of Rif1. Under the same conditions, loss of palmitoyl transferase Pfa4 caused a comparable decrease in cell survival (Fig. 1b), implicating Pfa4 in NHEJ. In line with previous observations[33], loss of Pfa4 did not affect Rif1 protein levels (Supplementary Fig. 1a). Interestingly, the NHEJ defect of pfa4Δ cells was not aggravated by the deletion of RIF1 (Fig. 1b). Disruption of NHEJ in yeast results in increased cell survival during chronic exposure to radiomimetic drugs such as Zeocin, consistent with HR being the more optimal pathway of DNA repair under these conditions[39]. We have shown previously that cells deleted for RIF1 exhibit a ~2-fold increase in Zeocin resistance compared to wild-type control cells[3]. A similar increase in survival upon Zeocin exposure was observed for cells deleted for PFA4, but not cells deleted for any of the other six palmitoyl-transferases present in budding yeast (Supplementary Fig. 1b). Furthermore, Zeocin resistance levels of rif1Δ pfa4Δ double-mutant cells were no greater than those of rif1Δ or pfa4Δ single mutant cells (Fig. 1c). These results suggest that Rif1 and Pfa4 may act jointly to facilitate NHEJ.

Across organisms, Rif1 promotes DSB repair by NHEJ through the attenuation of DNA end-resection[1]. To test whether Pfa4 impacts on DNA end-resection, we measured the accumulation of single-stranded DNA (ssDNA) at the induced DSB at the MAT locus using a qPCR-based method[40] (see Fig. 1a for details). Strikingly, loss of Pfa4 was associated with a ~2-fold increase in ssDNA (2 h time-point) after DSB induction, phenocopying the effect caused by loss of Rif1, while the combined loss of Rif1 and Pfa4 did not increase ssDNA accumulation further (Fig. 1d). Thus, like Rif1, Pfa4 is required to prevent hyper-resection at DSBs.

These findings suggest that protein S-palmitoylation is important for NHEJ efficiency in yeast, and implicate Pfa4 and Rif1 in a common pathway of DSB repair pathway choice.

**Rif1-mediated NHEJ is dependent on residues C466 and C473.** Having observed an epistatic defect in NHEJ efficiency after disruption of RIF1 and/or PFA4, we sought evidence that Pfa4 acts through Rif1 to promote NHEJ. To identify sites of potential Rif1 S-acylation involved in NHEJ, we performed a mutational analysis of Rif1 and screened for compromised NHEJ efficiency. This analysis was based on two assumptions: first, relevant S-acylated cysteine residues must be contained within the N-terminal domain of Rif1 (residues 1–1322, hereafter referred to as Rif1$_{NTD}$ for Rif1 N-terminal domain). This region of Rif1 was shown to be required and sufficient for promoting NHEJ, with cells expressing Rif1$_{NTD}$ from the endogenous RIF1 locus being as effective in promoting NHEJ as cells expressing full-length Rif1 (ref. [3]) (see also Supplementary Fig. 2a). Secondly, S-acylation must occur on cysteines likely to be surface-exposed, which we identified using the available crystal structure information for Rif1$_{NTD}$[3]. Of 19 cysteines present in Rif1, 14 are contained within

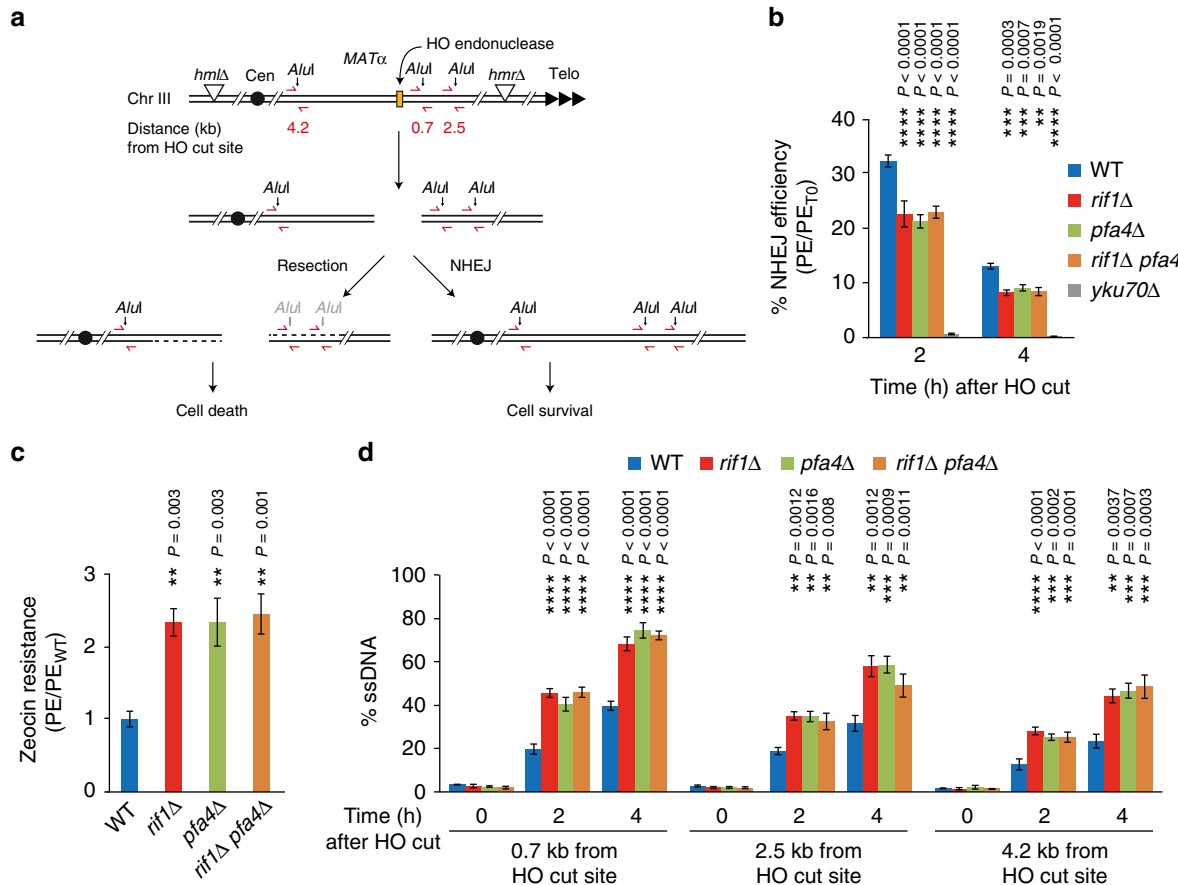

**Fig. 1** The palmitoyl acyltransferase Pfa4 promotes NHEJ. **a** Schematic representation of *S. cerevisiae* chromosome III harboring a *MATα* HO-endonuclease cut site. Upon DSB formation by induction of HO endonuclease, cell survival is possible by NHEJ while DNA end-resection leads to cell death due to lack of an HR repair template (*hml∆/hmr∆*). ssDNA formation progressively inactivates the annotated *Alu*I restriction sites, leading to increased qPCR product yield (primers indicated in red), providing a quantitative read-out for DNA end-resection. **b** NHEJ efficiency for the indicated strains measured by cell viability following 2 or 4 h of HO-endonuclease induction. Data are presented as mean values ± s.e.m. (*n* = 6 independent experiments). **c** Cell viability in the presence of Zeocin (70 μg/ml). Data are presented as mean values ± s.e.m. (*n* = 6 independent experiments). **d** ssDNA formed by DNA end-resection as determined by qPCR. Data are presented as mean values ± s.e.m. (*n* = 6 independent experiments). PE, plating efficiency; $PE_{T0}$, plating efficiency without HO-endonuclease induction; $PE_{RIF1}$, plating efficiency of *RIF1* wild-type reference strain; WT, wild-type. For statistical analysis, one-way analysis of variance (Anova) and a post-hoc Tukey–Kramer multiple comparison test was performed, comparing wild-type to the indicated mutants. See also Supplementary Fig. 1. Source data are provided as a Source Data file

$Rif1_{NTD}$, with C466, C473, C906, C1022, and C1089 being surface-exposed (Fig. 2a). In addition, we included in the analysis C71 and C1292, for which structural information is not available. Notably, in silico analysis using Swisspalm/CSS-Palm[41] predicted one of the cysteines that we selected, C466, as a residue of potential Rif1 *S*-palmitoylation (Supplementary Fig. 2b).

We grouped $Rif1_{NTD}$ residues C906, C1022, C1089, and C1292 (referred to as cluster 1), and C71, C466, and C473 (cluster 2) to facilitate the mutational analyses (Fig. 2a), targeting endogenous *RIF1* for alanine substitutions of all cysteines in cluster 1 and 2, alone and in combination, in the NHEJ reporter strain (see Fig. 1a). Single or combined alanine substitutions within cluster 1 had no effect on NHEJ efficiency as assessed by cell survival after DSB induction (Fig. 2b). In contrast, cluster 2 mutants Rif1 C71A/C466A/C473A and Rif1 C466A/C473A were associated with reduced survival after DSB induction, phenocopying *RIF1* (Fig. 2b) and/or *PFA4* (Fig. 1b) deletions. Single-site mutants Rif1 C466A or C473A had no effect on cell survival after DSB formation. Consistent results were obtained upon chronic exposure of cells to Zeocin, where the *rif1* C466A/C473A allele led to increased Zeocin resistance, similar to what is observed for *rif1∆*, *pfa4∆*, or *rif1∆ pfa4∆* cells, while the *rif1* C466A and *rif1*

C473A single-mutation alleles had no effect (Fig. 2c, see also Fig. 1c). All cluster 2 mutations had little or no impact on protein stability (Supplementary Fig. 2c). These results indicate an impairment of NHEJ by combined, but not individual loss of potential Rif1 *S*-acylation sites C466 and C473.

**Rif1 DSB end-protection and targeting requires C466 and C473.** To assess the functional consequences of mutating potential Rif1 *S*-acylation sites C466 and C473, we measured DNA end-resection using the inducible DSB at the *MAT* locus. Compared to cells expressing wild-type $Rif1_{NTD}$, cells expressing $Rif1_{NTD}$ C466A/C473A, but not those expressing $Rif1_{NTD}$ with a single C466A mutation, showed a ~2-fold increase of ssDNA accumulation (2 h time-point) (Fig. 3a, see also Supplementary Fig. 3a for NHEJ efficiency). This was not due to changes in protein levels (Supplementary Fig. 3b), nor a general loss of protein function since the $Rif1_{NTD}$ C466A/C473A mutant closely reflected the behavior of wild-type $Rif1_{NTD}$ in suppressing replication origin firing[12] (Supplementary Fig. 3c). The observed DSB hyper-resection associated with $Rif1_{NTD}$ C466A/C473A was comparable in cells deleted for *RIF1* or deleted for *PFA4*, and in

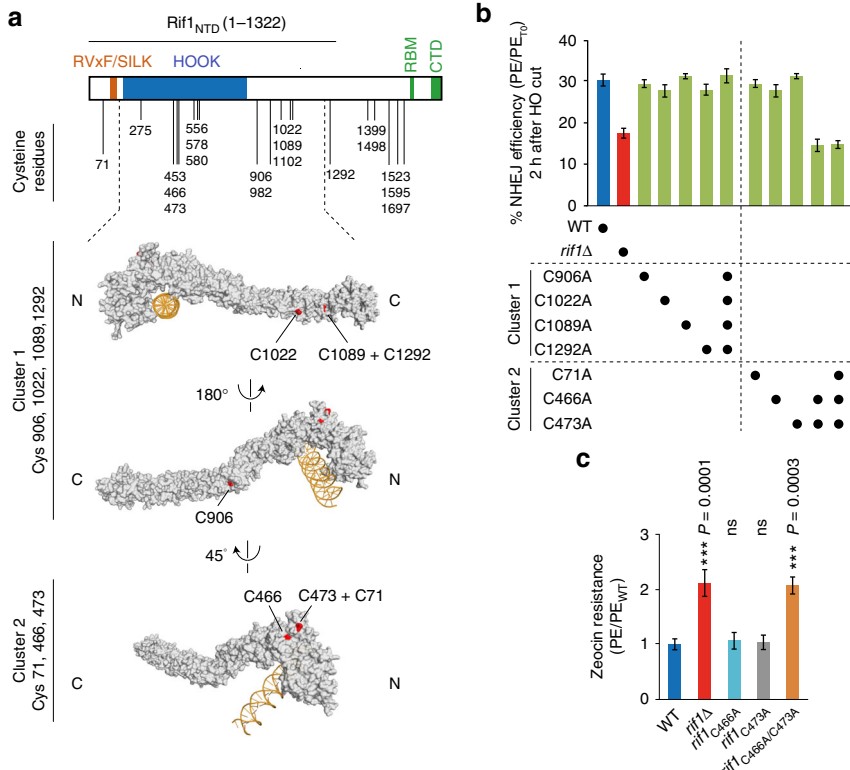

**Fig. 2** C466 and C473 are required for Rif1-mediated NHEJ. **a** Top, Budding yeast Rif1 (1916 amino acid residues) contains four identifiable functional domains: RVxF/SILK PP1-interacting domain; RBM and CTD Rap1-binding motifs; HOOK DNA-binding domain. The Rif1 N-terminal domain (Rif1$_{NTD}$, residues 1–1322) supports the protein's function in NHEJ. The positions of cysteine residues, representing potential Rif1 S-acylation sites, are indicated. Bottom, surface representation of Rif1 (residues 177–1283) structure bound with DNA[3], identifying surface-exposed cysteines. For mutational analyses, cysteines were grouped by proximity into clusters 1 and 2. C1292 and C71, for which structural data is not available, were included in cluster 1 and 2, respectively. **b** NHEJ efficiency for Rif1$_{NTD}$ cysteine clusters 1 and 2 mutants, determined as in Fig. 1b after 2 h of HO-endonuclease induction. Data are presented as mean values ± s.e.m. (n = 3 independent experiments). **c** Viability of the indicated strains in the presence of Zeocin (70 μg/ml). Data are presented as mean values ± s.e.m. (n = 6 independent experiments). Statistical analysis was performed by one-way Anova and a post-hoc Tukey–Kramer multiple comparison test, comparing wild-type to the indicated mutants. PE, plating efficiency; PE$_{T0}$, plating efficiency without HO-endonuclease induction; PE$_{RIF1}$, plating efficiency of RIF1 wild-type reference strain. See also Supplementary Fig. 2. Source data are provided as a Source Data file

cells expressing Rif1$_{NTD}$ DNA-binding site mutant K437E/K563E/K570E (referred to as HOOK mutant)[3] (Fig. 3a). Thus, like protective encapsulation of DNA ends by the HOOK domain, potential Rif1 S-acylation sites C466 and C473 are essential for Rif1-mediated DNA end-protection.

To analyze Rif1$_{NTD}$ occupancy at DSBs, we performed chromatin immunoprecipitation (ChIP) at the *MAT* DSB. As expected, Rif1$_{NTD}$ accumulated following cut induction, but occupancy compared to wild-type was reduced by ~40% (4 h time-point, 0.3 kb from DSB) when potential Rif1 S-acylation sites C466 and C473 were mutated, or when *PFA4* was deleted (Fig. 3b). Consistent with previous results[3], a similar loss of Rif1$_{NTD}$ from the DSB was observed upon introduction of the DNA-binding site HOOK mutation (Fig. 3b), and combining the HOOK and C466A/C473A mutations reduced Rif1$_{NTD}$ occupancy at DSBs further (Supplementary Fig. 3d, e). As shown in Fig. 2a, C466 and C473 are pointing away from the DNA-binding site, mapping to the convex surface of the HOOK domain. To rule out the possibility that the C466A/C473A mutation might disrupt DSB occupancy and cause NHEJ deficiency by disturbing the DNA-binding site at the concave surface of the HOOK domain, we purified Rif1 constructs (spanning residues 100–1322) with and without the C466A/C473A and HOOK mutations (Fig. 3c) to compare their DNA-binding activity using electromobility shift assays (EMSAs). As shown previously[3], the Rif1 DNA-binding site HOOK mutant K437E/K563E/K570E was

impaired in its ability to retard the DNA substrate in EMSAs. In contrast, the Rif1 C466A/C473A mutant exhibited a wild-type pattern of retarded DNA species, binding DNA with similar apparent affinity to wild-type (Fig. 3d). These results show that the potential Rif1 acceptor site C466/C473 for Pfa4-dependent S-acylation and the HOOK domain's DNA-binding site make separate contributions to Rif1's ability to effectively engage an induced DSB, while the integrity of both sites is indispensable for Rif1-mediated DNA end-protection and NHEJ.

**Pfa4-dependent S-acylation of Rif1 C466 and C473 in vivo.** To address the possibility of Rif1 S-acylation at C466 and C473, we turned to selective chemical labeling of thioester-linked cysteines. First, we used acyl-biotin exchange (ABE) followed by biochemical enrichment of S-acylated proteins. In brief, free cysteines are blocked with N-ethylmaleimide (NEM) before cysteine-acyl thioester cleavage using hydroxylamine (HA), with subsequent biotinylation at reactivated cysteines, allowing the capture of S-acylated proteins using avidin[42] (Fig. 4a). This procedure strongly enriched Rif1$_{NTD}$ from cell extracts in the avidin fraction compared to control reactions with HA omitted (Fig. 4b). Consistent with previous results[33], deletion of *PFA4* reduced the amount of Rif1$_{NTD}$ recovered by biotin-avidin affinity capture by ~5-fold. Importantly, capture of Rif1$_{NTD}$ C466A/C473A from extracts of Pfa4-proficient cells was more than 2-fold

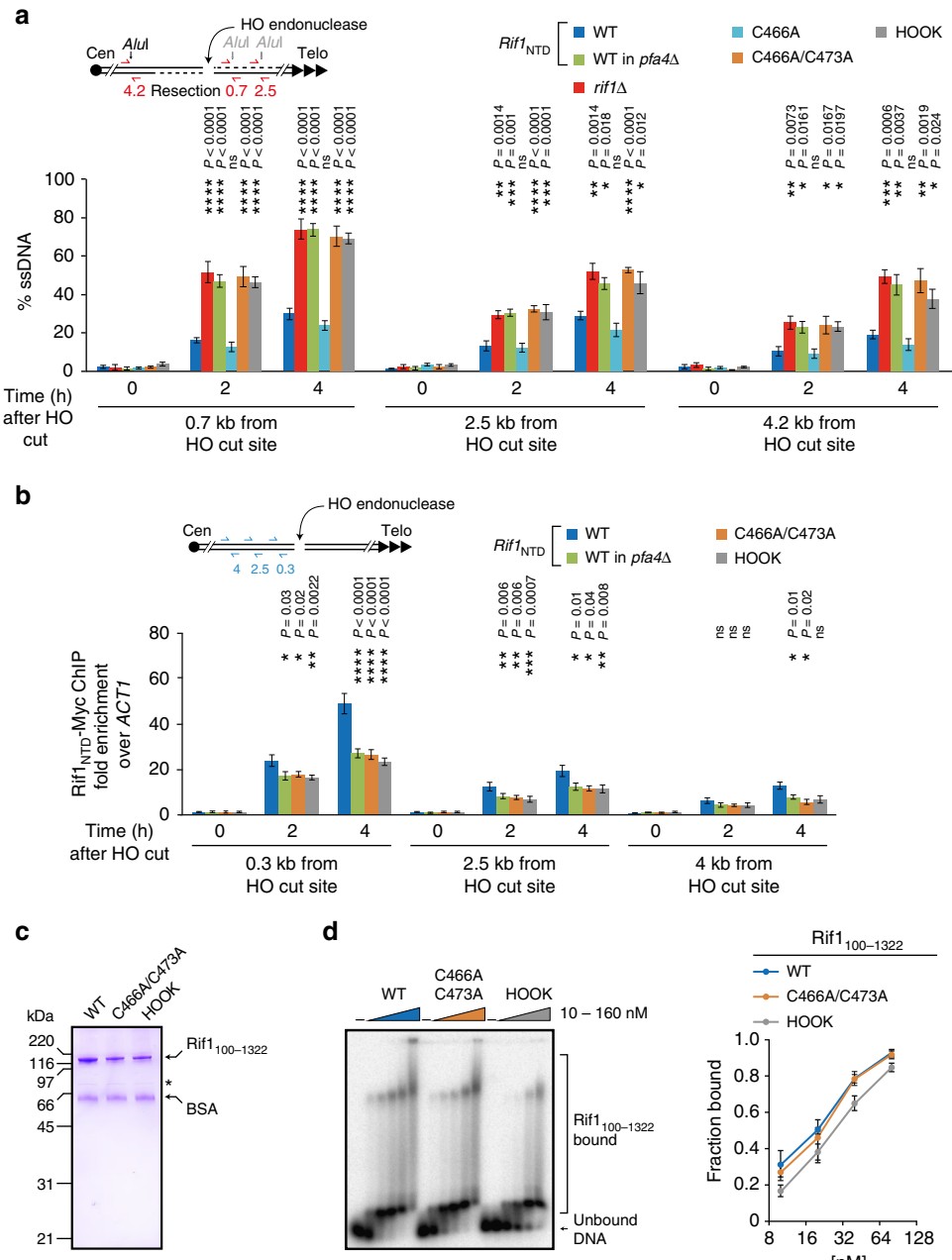

**Fig. 3** Rif1 C466 and C473 are required for DSB end-protection and targeting. **a** DNA end-resection upon DSB induction at the *MAT* locus in strains expressing wild-type (WT) Rif1$_{NTD}$, the indicated Rif1$_{NTD}$ mutants, or deleted for *RIF1*. ssDNA formation was determined by qPCR as in Fig. 1d. Data are presented as mean values ± s.e.m. (*n* = 6 independent experiments). **b** Association of wild-type and mutant Rif1$_{NTD}$-Myc with the DSB at *MAT*. Results obtained with the indicated primers are reported as fold enrichment relative to *ACT1* ± s.e.m. (*n* = 6 independent experiments). Statistical analyses shown in panels **a** and **b** were performed by one-way Anova, followed by a post-hoc Tukey–Kramer multiple comparison test, comparing wild-type to the indicated mutants. **c** SDS-PAGE gel of purified Rif1 (residues 100–1322) variants used for DNA-binding assays, stained with Coomassie blue. Asterisk denotes an unspecific band, BSA added for enhanced Rif1 stability. **d** EMSA analysis assessing the DNA-binding activity of the indicated Rif1$_{100-1322}$ variants (10–160 nM) using a $^{32}$P-labeled 30 bp dsDNA (1 nM). Left, representative EMSA with free and Rif1-bound DNA species indicated. Right, quantitation of bound DNA presented as mean values ± s.e.m. (*n* = 3 independent experiments). X-axis, log2 scale. See also Supplementary Fig. 3. Source data are provided as a Source Data file

reduced compared to wild-type Rif1$_{NTD}$ (Fig. 4b). These findings support NHEJ-critical Rif1 residues C466 and C473 as in vivo Rif1 *S*-acylation sites.

We next devised an alternative method to more directly establish Rif1 C466 and C473 *S*-acylation, which we term acyl-carbamidomethyl exchange (ACE). Different from ABE chemistry, ACE replaces fatty acylation with carbamidomethyl (CAM) rather than biotin and Rif1 is captured by immunoprecipitation,

such that modified peptides can be recovered, and *S*-acylation sites are accessible for mapping using mass spectrometry (Fig. 4c). For ACE, unmodified cysteines and tris (2-carboxyethyl) phosphine (TCEP)-reduced cysteines (opening potential disulfide bridges) were first labeled with NEM. This procedure leaves cysteine *S*-acylation intact (Supplementary Fig. 4a). Cysteine-acyl thioesters were then cleaved by treatment with dithiothreitol (DTT), followed by cysteine carbamidomethylation, allowing the

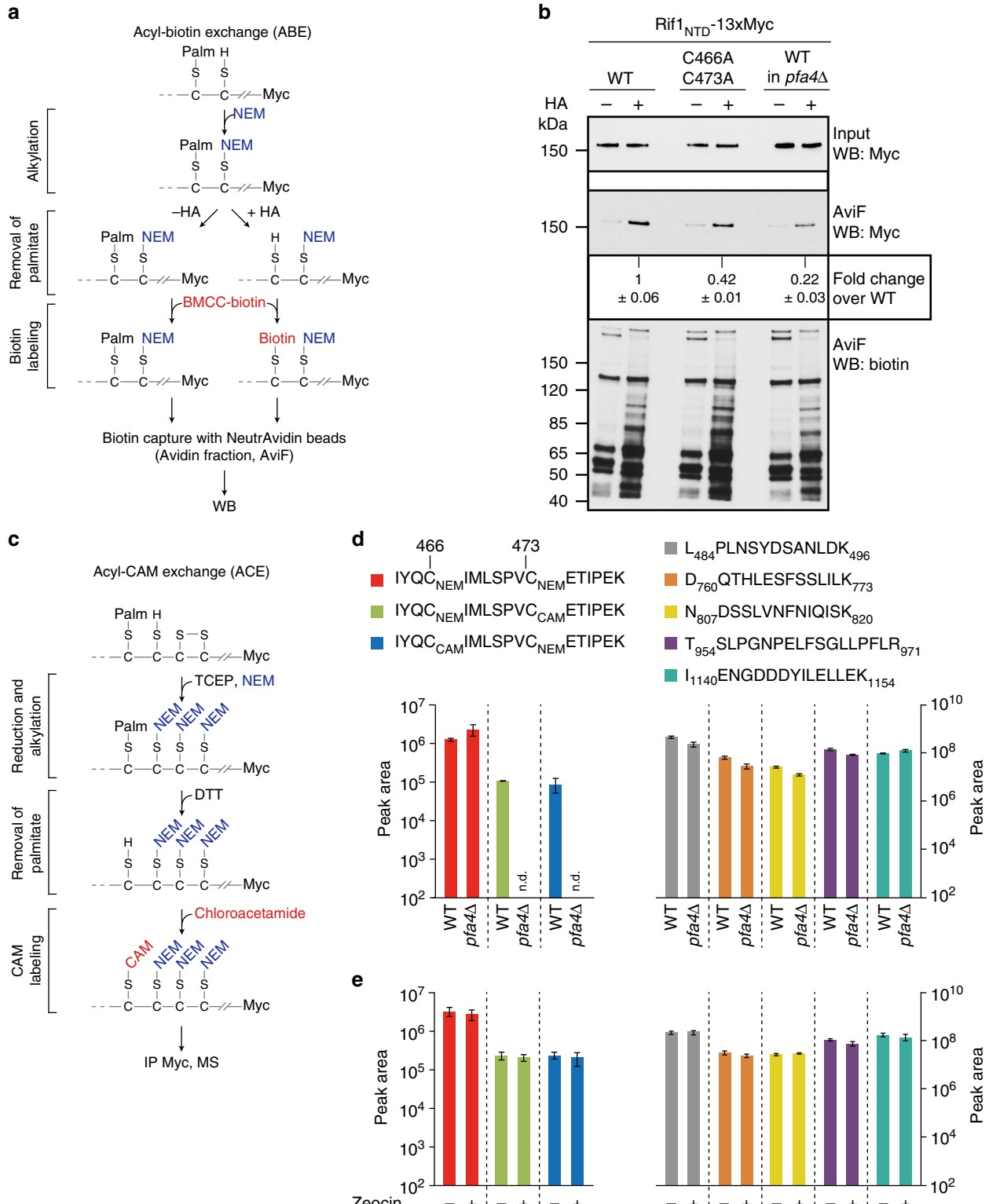

identification of *S*-palmitoylation sites in peptides with high sensitivity (see Supplementary Fig. 4b, c). Immuno-precipitated Rif1$_{NTD}$ was subjected to parallel reaction monitoring (PRM), analyzing NEM and/or CAM-labeled C466 and C473 in tryptic peptide fragments spanning residues 463 to 479. Peptides containing NEM-labeled C466 and C473 were detected in wild-type and *pfa4Δ* cells, accounting for unmodified Rif1$_{NTD}$. Tryptic fragments containing either CAM-labeled C466 or CAM-labeled C473 were detected in wild-type cells, providing site-specific

evidence for Rif1 *S*-acylation in vivo (Fig. 4d). Under these conditions, CAM-modified Rif1 peptides from *pfa4Δ* cells were not detected (Fig. 4d, see also Supplementary Fig. 4 and Supplementary Table 1 for further peptide analyses), supporting the role of Pfa4 in Rif1 C466/C473 *S*-acylation. Alternative *S*-acylation at C466 and C473 might explain why only combined Rif1 C466A/C473A mutations result in defective NHEJ, suggesting site redundancy for NHEJ-relevant *S*-acylation by Pfa4 in vivo.

**Fig. 4** *S*-acylation of Rif1 C466 and C473 in vivo. **a** Outline of the ABE protocol, including protein alkylation at free cysteines with NEM, removal of *S*-acyl groups including palmitoyl (Palm) using hydroxylamine (HA), and labeling with BMCC-biotin. Biotinylated proteins are captured on NeutrAvidin-coated beads and the presence of Rif1$_{NTD}$ is analyzed by western blotting (WB). **b** Representative western blots of ABE assays performed with cells expressing Myc-tagged Rif1$_{NTD}$ under control of a *GAL1* promoter. Input: BMCC-biotin samples prior to biotin capture with and without HA. Input and AviF samples were probed with anti-Myc and anti-biotin antibodies as indicated. Fold enrichment of Rif1$_{NTD}$ in AviF relative to wild-type is presented as mean values ± s.e.m. ($n = 3$ independent experiments). **c** Outline of the ACE protocol, including treatment of proteins with TCEP (reducing potential disulfide bridges between cysteine residues), alkylation at free cysteines with NEM, removal of *S*-acyl groups using DTT, and alkylation of freed-up cysteines with chloroacetamide. Myc-tagged Rif1$_{NTD}$ is then immunoprecipitated for analysis by mass spectrometry (see Supplementary Fig. 4a for additional controls). **d** Mass-spectrometric analysis of tryptic Rif1 fragments spanning amino acids 463 to 479. Following ACE, Rif1$_{NTD}$ tryptic peptides were subjected to parallel reaction monitoring (PRM), measuring NEM (unmodified Rif1) and/or CAM-labeled (reflecting in vivo *S*-acylation) C466 and C473 in wild-type vs. *pfa4Δ*. Integrated PRM counts are presented as mean values ± s.e.m. and were normalized using measurements of the five non-modified Rif1 peptides shown on the right (see Supplementary Table 1 for additional information). Data are shown in logarithmic scale ($n = 3$ independent experiments). **e** Measurements of C466/C473 NEM and/or CAM-labeled peptides (left panel) and unmodified control peptides (right panel) of Rif1 in untreated vs. Zeocin-treated wild-type cells. PRM analysis of Rif1$_{NTD}$ peptides as in panel **d**. Mean values of integrated PRM counts ± s.e.m ($n = 3$ independent experiments) are shown in logarithmic scale. See Supplementary Fig. 4b, c for peptide transitions used in the experiment. Source data are provided as a Source Data file

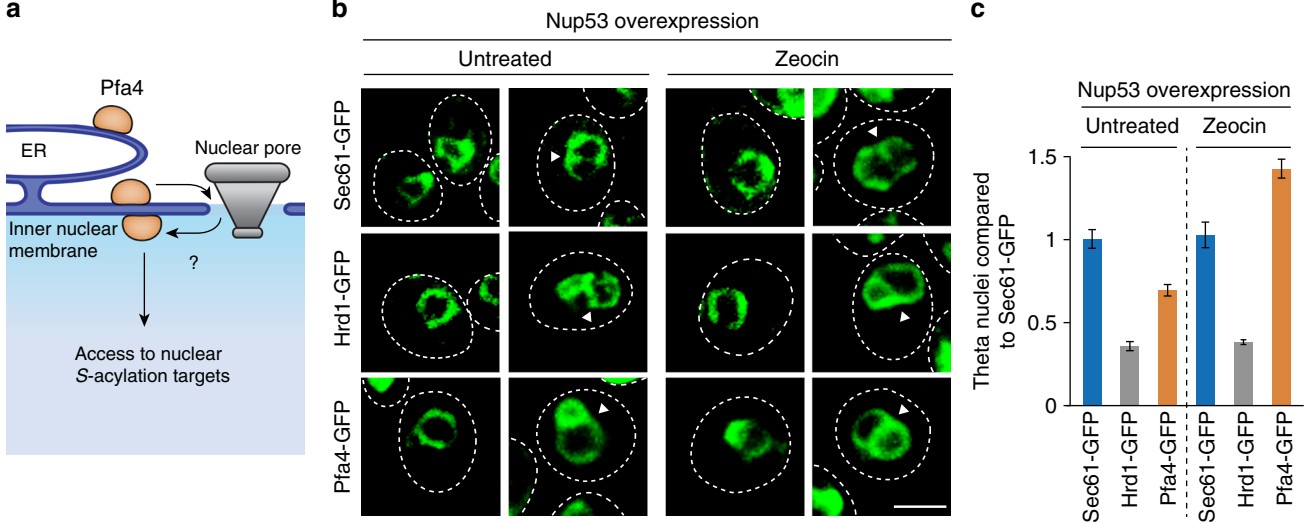

**Fig. 5** Pfa4 associates with the inner nuclear membrane. **a** Pfa4 localizes to the endoplasmic reticulum (ER)[43] but access to the inner nuclear membrane has not been demonstrated. **b** Cells with theta (*θ*) nuclei, induced by Nup53 overexpression-mediated inner nuclear membrane proliferation, were analyzed for inner nuclear membrane-association of Sec61-GFP, Hrd1-GFP, and Pfa4-GFP (arrowheads indicate GFP-positive *θ* structures). Representative Z-projections of confocal microscopy images are shown for cells treated or not with Zeocin. Scale bar: 5 μm. **c** Quantification of the analysis shown in panel **b**. Results from three independent experiments ($n ≥ 400$ total cells per strain and condition) are shown as mean values ± s.e.m.

The PRM values for double NEM-modified Rif1 peptides spanning residues 463 to 479 are ~10 times higher than the values for peptides derived from Rif1 *S*-acylated at C466 or C473. This implies that a substantial fraction of 15–20% of Rif1 was *S*-acylated at either C466 or C473 in wild-type cells (Fig. 4d). DNA-damage treatment with Zeocin did not lead to gross changes in Rif1 *S*-acylation levels (Fig. 4e), consistent with constitutive Rif1 *S*-acylation.

**Rif1 S-acylation allows a nuclear-peripheral damage response.** To address the question how *S*-acylation may promote Rif1-mediated NHEJ, we first sought to determine whether the modifying enzyme, Pfa4, has access to the cell nucleus. Although membrane associated, palmitoyl transferases have so far not been observed at the inner nuclear membrane. Pfa4-GFP has been localized at the endoplasmic reticulum (ER)[43], which is continuous with the nuclear envelope, but inner nuclear membrane access is selective, and whether Pfa4 can populate this subcompartment is unknown (Fig. 5a). Taking advantage of induced inner nuclear membrane proliferation following overexpression of nucleoporin Nup53, we asked whether Pfa4-GFP can access

the resulting, distinctive membrane structures. These intranuclear lamellae have been shown to present in the form of so-called theta (*θ*) nuclei with transecting membranes, providing the basis for a quantifiable fluorescence assay for testing inner nuclear membrane localization of GFP-tagged candidate proteins[44]. As a positive control, we expressed ER membrane protein Sec61-GFP, which accessed the inner nuclear membrane, efficiently decorating *θ* structures[44] induced by Nup53 overexpression (Fig. 5b). As expected, cells expressing ER membrane protein Hrd1-GFP showed ~4-fold lower levels of fluorescent *θ* nuclei compared to Sec61-GFP (Fig. 5c), reflecting poorer inner nuclear membrane access[44]. Pfa4-GFP showed an intermediate phenotype, populating *θ* nuclei at ~1.5-fold lower levels than Sec61-GFP (Fig. 5c). Upon DNA-damage treatment with Zeocin, the localization of Sec61-GFP or Hrd1-GFP to *θ* structures did not change, while Pfa4-GFP-associated *θ* structures increased ~2-fold (Fig. 5b, c). These data show that Pfa4 localizes to the inner nuclear membrane in unperturbed conditions, and this localization is enhanced after DNA damage. Using cell fractionation, we confirmed previous results[33] indicating Pfa4-dependent membrane associations of Rif1, which proved partially dependent on Rif1 residues C466 and C473 (Supplementary Fig. 5). Together, these

results are consistent with Pfa4 having access to Rif1 in the nucleus, where NHEJ-relevant *S*-acylation at C466 and C473 may contribute to Rif1-membrane interactions.

To investigate whether *S*-acylation-mediated membrane anchorage may direct the actions of Rif1 in NHEJ to the inner nuclear membrane, we expressed fluorescently tagged Rif1$_{NTD}$, which is unable to interact with Rap1 (ref. [2]) and does not co-localize with telomere clusters (Supplementary Fig. 6a). In untreated conditions, we observed nuclear Rif1$_{NTD}$-GFP foci in ~28% of wild-type cells. After DNA-damage treatment with Zeocin or ionizing radiation (IR), focus formation was strongly induced, reaching a peak ~30 min post-treatment (Supplementary Fig. 6b), when ~60% (Zeocin) and ~80% (IR) of cells exhibited Rif1$_{NTD}$-GFP foci (Fig. 6a, b and Supplementary Fig. 6c, d). Moreover, while the majority of focus-positive cells in unperturbed conditions contained a single Rif1$_{NTD}$-GFP focus, most

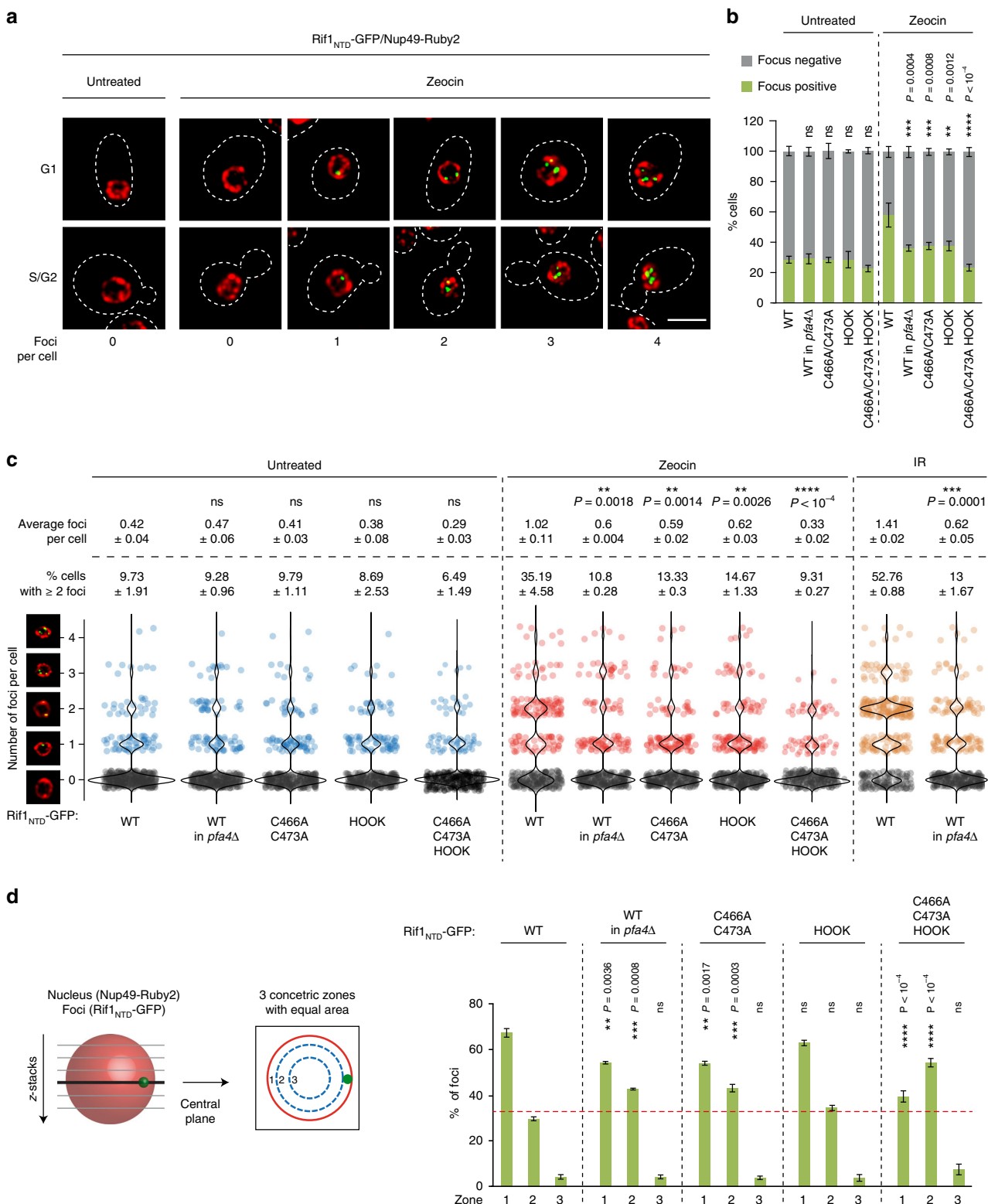

**Fig. 6** Rif1 C466/C473 *S*-acylation mediates a peripheral DNA damage response. **a** Confocal microscopy of cells expressing Rif1$_{NTD}$-GFP and Nup49-Ruby2, untreated or treated with Zeocin (100 μg/ml, 30 min; see Supplementary Fig. 6c, d for IR treatment). Z-projected images of cells in the indicated cell cycle phases, illustrating the observed classes of focus-negative and Rif1$_{NTD}$ focus-positive (1–4 foci) cells. For untreated cells, only the main class (focus-negative, representing ~75% of cells) is shown. Scale bar: 5 μm. **b** Quantification of focus-positive and focus-negative cells for the indicated strains expressing wild-type or mutant Rif1$_{NTD}$-GFP, treated or not with Zeocin. Results of three independent experiments ($n \geq 100$ cells per experiment) are presented as mean values ± s.e.m. For statistical analysis of GFP-positive cells treated or not with Zeocin, one-way Anova and a post-hoc Tukey–Kramer multiple comparison test was performed, comparing wild-type to mutants. **c** Quantitation of Rif1$_{NTD}$-GFP foci per cell for the indicated strains with and without DNA-damage treatment (Zeocin, IR). Violin plots show data from three independent experiments ($n \geq 95$ cells per strain and condition), binned for number of foci per cell. The average number of foci per cell (± s.e.m.) and the fraction of cells with $\geq 2$ foci (± s.e.m.) are indicated. For statistical analysis, comparing wild-type to the indicated mutants with and without DNA-damage treatment, one-way Anova and a post-hoc Tukey–Kramer multiple comparison test was performed; unpaired t-test for cells treated or not with IR. **d** Zoning assay scoring the position of Rif1$_{NTD}$-GFP foci in the indicated strains treated with Zeocin relative to the nuclear envelope (marked by Nup49-Ruby2). Foci were binned in three concentric zones of equal area; dashed line at 33% indicates random distribution. Results of three independent experiments ($n = 50$ Rif1 foci from 50 focus-positive cells per strain per experiment) are presented as mean values ± s.e.m. For statistical analysis, one-way Anova and a post-hoc Tukey–Kramer multiple comparison test was performed. Nuclear zones in wild-type were compared to the corresponding zones in the indicated mutants. See also Supplementary Fig. 6. Source data are provided as a Source Data file

focus-positive cells exhibited multiple (up to four) foci after DNA damage treatment (Fig. 6c). DNA damage-induced Rif1$_{NTD}$-GFP foci were observed in G1 and S/G2 cells with no overt cell-cycle dependence. DNA-damage treatment did not lead to increased Rif1$_{NTD}$ expression levels (Supplementary Fig. 6e), suggesting that focus formation reflected the redistribution of Rif1$_{NTD}$-GFP into foci upon DNA damage.

Next, we analyzed Rif1$_{NTD}$-GFP foci in *pfa4Δ* cells. Untreated cells were indistinguishable from wild-type with ~30% Rif1$_{NTD}$-GFP focus-positive cells, the majority of which contained a single Rif1$_{NTD}$-GFP focus (Fig. 6b, c and Supplementary Fig. 6c, d). In contrast, the ability to form DNA damage-induced Rif1$_{NTD}$-GFP foci (~36% and ~38% focus-positive cells after Zeocin and IR, respectively, see Fig. 6b and Supplementary Fig. 6d, f) and the formation of multiple Rif1 foci in response to DNA damage (Fig. 6c) was significantly abrogated in *pfa4Δ* cells. Protein levels of Rif1$_{NTD}$ remained unchanged upon loss of Pfa4 (Supplementary Fig. 6g). Importantly, in Pfa4-proficient cells, introducing the *S*-acylation mutation C466A/C473A also diminished the formation of DNA damage-induced Rif1$_{NTD}$ foci and the ability of cells to form multiple Rif1 foci (Fig. 6b, c). Like *S*-acylation mutant Rif1$_{NTD}$ C466A/C473A, the Rif1 HOOK DNA-binding mutant was strongly compromised in its ability to form foci in response to DNA-damage treatment (Fig. 6b, c). Combining the *S*-acylation and HOOK mutations led to a more severe phenotype compared to either the Rif1$_{NTD}$ C466A/C473A or the Rif1 HOOK mutant (Fig. 6b, c). Thus, Pfa4-dependent *S*-acylation of Rif1 at C466/C473 and the ability of Rif1 to bind DNA contribute to effective Rif1 accumulation upon DNA damage.

To determine the sub-nuclear localization of Rif1$_{NTD}$-GFP foci, we scored their position relative to the nuclear envelope marked by fluorescently tagged nuclear-pore component Nup49 (ref. [45]). Dividing the nucleus into three concentric zones of equal area, we found a strong bias of Rif1 accumulation in outermost zone 1, at the nuclear periphery, in wild-type cells. Upon DNA damage, ~60% of Rif1$_{NTD}$-GFP foci localized in zone 1 (Fig. 6d). While cells expressing the Rif1 HOOK DNA-binding mutant maintained a strong localization bias to zone 1, cells with compromised Rif1 *S*-acylation expressing Rif1$_{NTD}$ C466A/C473A, Rif1$_{NTD}$ C466A/C473A HOOK, or wild-type Rif1$_{NTD}$ in a *pfa4Δ* background, exhibited an increase in zone 2-localized foci at the expense of zone 1-localized foci. Thus, Rif1 *S*-acylation mutants display an apparent reduction in Rif1-inner nuclear membrane interactions in conjunction with a significant impairment in the formation of DNA damage-induced Rif1-foci observed in all mutant backgrounds tested (Fig. 6b, c and Supplementary Fig. 6c, d, f).

Taken together, these data are consistent with a model where enrichment of Rif1 at the inner nuclear membrane mediated by Pfa4-dependent *S*-acylation of C466/C473 and its intrinsic DNA-binding activity enable effective Rif1 accumulation at nuclear-peripheral DNA damage, promoting preferential repair of membrane-proximal DSBs along the NHEJ pathway (Fig. 7).

## Discussion

Here, we show that the palmitoyl transferase Pfa4 is essential for Rif1-dependent DSB repair by NHEJ. Loss of Pfa4 caused a drop in NHEJ efficiency but did not exacerbate the NHEJ defect of *rif1Δ* cells, genetically implicating Pfa4 and Rif1 in a common DSB repair pathway (Fig. 1). Starting from a structure-guided approach, we identified Rif1 C466 and C473 as two cysteine residues required for efficient NHEJ in vivo (Fig. 2), accumulation of Rif1 at DSBs, and attenuation of DNA end-resection (Fig. 3). We demonstrate that C466 or C473 are modified by *S*-acylation in vivo in a strictly Pfa4-dependent manner, strongly suggesting they are direct Pfa4 *S*-palmitoylation targets (Fig. 4). Residual avidin-enrichment after subjecting Rif1$_{NTD}$ C466A/C473A to ABE chemistry points to Rif1 *S*-acylation at additional, as-yet unmapped sites (Fig. 4b). While these potential additional sites are neither required nor sufficient for Rif1 to promote NHEJ, at least one Pfa4-dependent *S*-acylation event at C466 or C473 is essential for Rif1-mediated NHEJ (Fig. 2). Localizing to the inner nuclear membrane (Fig. 5), Pfa4 has access to nuclear Rif1 and promotes Rif1-membrane associations[33] (Supplementary Fig. 5). Furthermore, we discovered the nuclear-peripheral, focal accumulation of Rif1$_{NTD}$ in response to treatment of cells with radiomimetic drug Zeocin or IR. This localized DNA-damage response of Rif1 occurred in cells in G1 and S phase of the cell cycle, and thus reflects a cell-cycle phase-independent process. Importantly, DNA damage-induced Rif1 focus formation was dependent on Pfa4, *S*-acylated Rif1 residues C466 and C473, and required Rif1's DNA-binding activity (Fig. 6). As IR and Zeocin give rise to DSBs and ssDNA breaks, we cannot exclude the possibility that Rif1 responds to multiple types of DNA damage. However, Rif1 accumulates at endonuclease-induced DSBs in a Pfa4 and C466/C473-dependent manner to attenuate DNA end-resection and promote NHEJ (Fig. 3 and Supplementary Fig. 3a). We therefore propose that the focal accumulation of Rif1 at the nuclear periphery, induced by Zeocin or IR, is a direct reflection of DSB binding. This contrasts with HR-dependent DSB repair foci marked by Rad52, which are generally found within the nuclear lumen[46], a compartment that was deprived of DNA damage-induced Rif1 foci. Based on our results, we suggest a model where posttranslational fatty acylation at C466 or C473 increases the affinity of Rif1 for the inner nuclear membrane, creating a nuclear-peripheral compartment of high local Rif1 concentration. Here, Rif1 is poised to mount an effective response

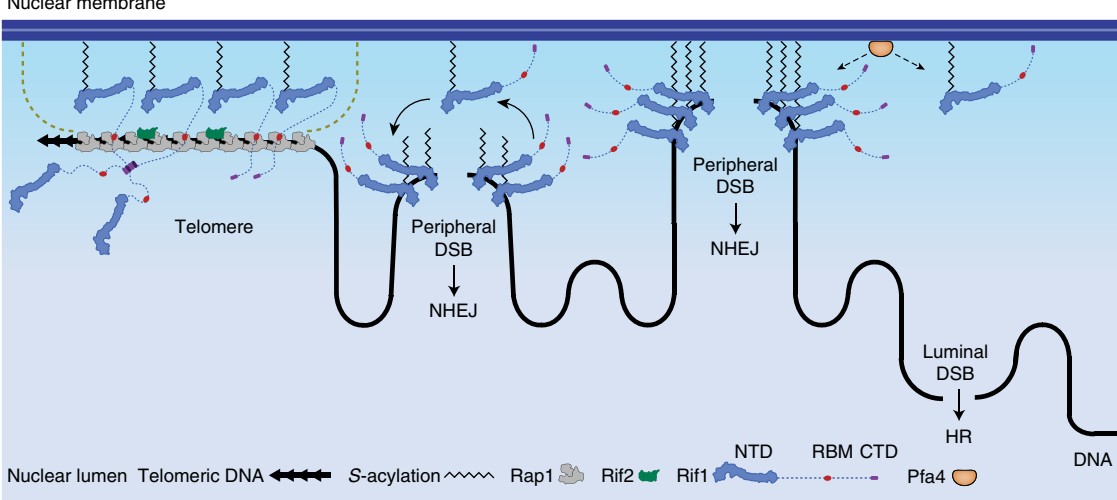

**Fig. 7** Rif1 *S*-acylation mediates nuclear-peripheral DSB repair pathway choice. Pfa4-dependent *S*-acylation of Rif1 provides membrane anchors and governs localization of Rif1 to the inner nuclear membrane. Telomeres are tethered to the nuclear envelope by Rif1-independent mechanisms (dashed lines) and Pfa4 is dispensable for telomere homeostasis. In contrast, Pfa4-dependent Rif1 *S*-acylation at residues C466 or C473 (depicted as a zig-zag line) is essential for Rif1-mediated NHEJ. We propose that the increased local concentration of *S*-acylated Rif1 in proximity of the inner nuclear membrane generates a nuclear-peripheral compartment in which Rif1 is effective in promoting DSB repair by NHEJ

to DNA damage, encapsulating DSBs with its HOOK domain, and dampening DNA end-resection to favor repair by NHEJ (Fig. 7).

It is tempting to speculate that membrane binding may provide Rif1 with additional means of promoting favorable DNA repair outcomes, for example by constraining the movement of DSB ends to suppress chromosome instability by ectopic recombination events. At the same time, membrane-attachment could allow Rif1 to harness the nuclear envelope as a scaffold to facilitate the coordination of DSB ends for re-ligation. For telomere maintenance, Rif1 *S*-acylation appears to be dispensable[33], and this may be explained by the recruitment of Rif1 to telomeres through protein-protein interactions with Rap1 (ref. [2]). However, *S*-acylation may strengthen telomere interactions at the nuclear periphery[33] to reinforce the telomere position effect[33], where Rif1 helps antagonize transcriptional silencing near telomeres[47,48]. Rif1 interactions with the nuclear envelope have also been implicated in chromatin architecture and DNA replication[5,7,14,15,49,50]. While the ability of Rif1 to suppress DNA replication origin firing does not depend on Pfa4 (ref. [8]) or C466 and C473 (Supplementary Fig. 3c), it will be important to explore at high resolution whether Rif1 *S*-acylation mutants act at ectopic genomic sites to determine whether fatty acylation modulates replication timing programs in eukaryotes.

The link between Rif1 and nuclear-peripheral DNA repair described herein resonates with increasing evidence for DNA repair compartmentalization and the impact of DSB microenvironments on repair pathway choice across organisms[51,52]. In human cells, nuclear-peripheral DSBs bound at the proteinaceous nuclear lamina are preferentially repaired by NHEJ[53]. The compaction of silent heterochromatin and repetitive sequence elements in lamina-associated domains has been proposed to hamper the recruitment of HR proteins, leading to a local NHEJ bias[53]. Reported lamina interactions of Rif1 (refs. [5,7,49]) raise the interesting possibility that peripheral sequestration of Rif1 might have a role in sub-nuclear compartments geared towards NHEJ in mammalian cells. Of note, mammalian Rif1 *S*-palmitoylation is predicted by Swisspalm/CSS-Palm[41], but it remains to be determined whether these modifications occur in vivo and how they might relate to NHEJ. In yeast, *S*-acylated Rif1 may preferentially target DSBs for NHEJ in nuclear

envelope-associated heterochromatin or near telomeres. Furthermore, persistent DSBs in yeast relocate to the nuclear periphery[54–56], which may facilitate access by *S*-acylated Rif1.

Protein *S*-acylation has been detected on a wide range of cytosolic and nuclear proteins[31,57], and dysfunctional fatty acylation has been implicated in human diseases including cancer[58,59]. However, the functional consequences for most targets of *S*-acylation remain to be determined. Interestingly, chemical inhibition of protein *S*-palmitoylation in mammalian cells led to a muted DNA-damage response[60]. Here, we uncover a role for protein *S*-acylation in DSB repair pathway choice. The reversible nature of *S*-palmitoylation is reminiscent of well-established posttranslational modifications with important roles in DSB repair, including protein phosphorylation and ubiquitination, potentially allowing the dynamic regulation of Rif1 at DNA damage. The essential requirement for *S*-acylation in Rif1-mediated NHEJ provides the first example of a direct involvement of fatty acylation in DNA repair.

## Methods

**Yeast techniques.** The complete list of *S. cerevisiae* strains used in this study can be found in Supplementary Table 2. Deletions and epitope tagging of genes of interest were done by one-step PCR gene replacement[61]. Point mutations in the *RIF1* gene were introduced using delitto perfetto[62] and/or CRISPR/Cas9-based methods[63] (see Supplementary Table 3 for primer sequences). For overexpression of Myc-tagged Rif1$_{NTD}$, Pfa4-GFP, and Nup53, a *GAL1* galactose-inducible promoter was inserted genomically and expression was induced by addition of 2% (w/v) galactose (Formedium, GAL02) to cells cultured in YPLG. For drop assays, strains were grown exponentially, and serial 10-fold dilutions were spotted on YPAD agar plates before incubation at the indicated temperatures as described previously[12]. Plates were imaged after 2 to 3 days.

**NHEJ assays.** The efficiency of NHEJ as measured by cell survival was determined as described[38,64]. JKM179-derived strains (see Supplementary Table 2) were grown overnight in YPAD, then diluted in YPLG and grown exponentially. For transient HO-endonuclease expression, 2% (w/v) galactose was added to the culture medium. At the indicated time-points, cells were plated on glucose-containing YPAD agar plates. Control cells were removed prior to HO-endonuclease induction and plated on medium containing glucose. HO endonuclease cut-efficiency was routinely determined by qPCR (see Supplementary Table 3 for primer sequences) and data was normalized accordingly[3]. Colonies were counted 3 days after plating, and NHEJ efficiency was calculated as described[64].

**Determination of cell viability in the presence of Zeocin.** The plating efficiency of each strain was determined on drug-free medium. Exponentially growing cells were diluted appropriately and plated on YPAD medium containing 70 μg/ml Zeocin (ThermoFisher Scientific, R25001). Colony outgrowth was quantified after incubation for 3–4 days at 30 °C. Zeocin resistance relative to wild-type cells was determined as described[3].

**Quantification of ssDNA as a measure of DNA end-resection.** Quantification of ssDNA was done as described previously[3]. In brief, genomic DNA was extracted using a phenol:chloroform and propan-2-ol method, then digested overnight with *AluI* (New England Biolabs, R0137L). qPCR was performed using GoTaq qPCR Master Mix (Promega, A6001). To detect the formation of ssDNA, primer pairs flanking *AluI* sites in the proximity of an HO endonuclease-induced DSB were used. *AluI* sites that have been converted to ssDNA by end-resection are resistant to cleavage, leading to increased qPCR yield[40]. To normalize the data, qPCR amplifications of genomic regions devoid of *AluI* sites (located at the *SMC2* locus) were performed. qPCR reactions were carried out to assess the efficiency of DSB-induction by the HO endonuclease. Primer sequences are reported in Supplementary Table 3.

**Western blotting.** Western blots were performed as described[12], using the following primary antibodies: mouse monoclonal anti-beta actin (Abcam, ab8224, RRID: AB_449644, 1:1000 dilution), rat monoclonal anti-tubulin (Abcam, ab6161, RRID: AB_305329, 1:4000 dilution), mouse monoclonal anti-Pgk1 (Abcam, ab113687, RRID: AB_10861977, 1:5000 dilution), rabbit polyclonal anti-biotin (Abcam, ab1227, RRID: AB_298990, 1:3000 dilution), mouse monoclonal anti-c-Myc (clone 9E10, Sigma-Aldrich, M4439, RRID: AB_439694, 1:4000 dilution), mouse monoclonal anti-nuclear pore complex proteins (Mab414, Abcam, ab24609, RRID: AB_448181, 1:5000 dilution). Secondary antibodies used were goat anti-rat IgG (Abcam, ab97057, RRID: AB_10680316, 1:10,000 dilution), donkey anti-rabbit IgG (GE Healthcare, GENA934, RRID: AB_2722659, 1:10,000 dilution) and sheep anti-mouse IgG (GE Healthcare, NA931, RRID: AB_772210, 1:10,000 dilution). See "Quantification and statistical analysis" for a detailed description of western blot quantifications.

**ChIP assays.** To detect enrichment of proteins in proximity of the HO-endonuclease cut-site at the *MATa* locus, cells were grown logarithmically in YPLG medium and the HO endonuclease was then induced by adding 2% (w/v) galactose. ChIP was performed as reported previously[3] (see Supplementary Table 3 for primer sequences).

**Rif1$_{NTD}$ protein purification.** pFastBac-derived constructs of the wild-type and mutant *S. cerevisiae* Rif1 N-terminal domain spanning residues 100–1322 were expressed as N-terminal Strep(II)-tag fusions in *Trichoplusia ni* High Five insect cells (ThermoFisher Scientific, B85502). Bacmids, primary and secondary viruses were produced using the Bac-to-Bac baculovirus expression system (ThermoFisher Scientific, 10359–016) and *Spodoptera frugiperda* (Sf9) insect cells (ThermoFisher Scientific, 11496015), as described previously[3]. Insect cell expression cultures of Rif1$_{NTD}$ (residues 100–1322) and Rif1$_{NTD}$ mutants (HOOK DNA-binding mutant: K437E/K563E/K570E; *S*-acylation mutant: C466A/C473A) were grown in SF900-II medium (ThermoFisher Scientific, 10902096) at 27 °C and infected at a cell density of $4 \times 10^6$ ml with 15 ml/l of P2 virus solution. Cells were harvested by centrifugation 48 h after infection and lysed by sonication in 50 mM Tris-HCl pH 8.0, 500 mM NaCl, 1 mM TCEP, 1 mM PMSF, 5 mM 2-mercaptoethanol, and protease-inhibitor cocktail (Sigma, S8830–20TAB). After clarification by ultracentrifugation (45,000×*g* for 45 min at 4 °C), Rif1$_{NTD}$ was affinity-extracted using Strep-Tactin sepharose beads (IBA, 2–1201–010). The protein solution was concentrated by ultrafiltration (Amicon Ultra-15 centrifugal filter unit, 30 KDa cutoff, Merck, UFC903024) and separated by size exclusion chromatography using HiLoad 26/600 Superdex 200 pg (GE Healthcare, 28989336) in 50 mM HEPES pH 7.4, 310 mM NaCl, 1 mM TCEP. Purified Rif1$_{NTD}$ was concentrated by ultrafiltration once again, flash-frozen in liquid nitrogen, and stored at −80 °C.

**DNA-binding assays.** The double-stranded DNA substrate for Rif1 EMSAs[3] was assembled from polyacrylamide gel electrophoresis (PAGE)-purified oligonucleotides EMSA1 and EMSA2 (Microsynth AG, see Supplementary Table 3 for sequences). EMSA1 was labeled using T4 polynucleotide kinase (New England BioLabs, M0201L) and [γ-$^{32}$P] ATP (Hartmann Analytic, SRP-301) at 37 °C for 1 h. The reaction was stopped by adding 20 mM EDTA and EMSA1 was purified using an Illustra MicroSpin G-25 column (GE Healthcare, 27532501) into 10 mM Tris-HCl pH 8.0, 1 mM MgCl$_2$, 50 mM NaCl. EMSA 1 and 2 were mixed at 2 μM concentration and annealed (in 10 mM Tris-HCl pH 8, 1 mM MgCl$_2$, 50 mM NaCl) by heating for 5 min to 98 °C and cooling at 1 °C/min to 4 °C. The annealed DNA substrates (20 μl) was purified by 14% PAGE electrophoresis as described[65] and stored at 4 °C in 10 mM Tris-HCl pH 8.0, 1 mM EDTA. Rif1$_{NTD}$ protein concentrations were measured using the Bradford method with a bovine serum albumin (BSA) standard. EMSAs were performed as described previously[3]. In brief, the labeled DNA substrate at a final concentration of 1 nM was added to serially diluted protein to obtain the protein concentrations indicated (10–160 nM) in a buffer containing 20 mM Tris-HCl pH 8.0, 100 mM NaCl, 2.5 mM MgCl$_2$, 10 mM CaCl$_2$, 0.1 mg/ml BSA, 1 mM TCEP. The mixture was incubated for 30 min at 20 °C before adding glycerol at a final concentration of 8% (v/v), and separating a

10-μl sample by 1.2% agarose gel electrophoresis at 150 V for 2 h, in 0.5× TBE at 4 °C. Gels were dried on DE81 Whatman chromatography paper (Sigma-Aldrich, Z286591), exposed to storage phosphor screens (BioRad, 1707843), scanned by a Typhoon phosphorimager, and quantified as reported below in section "Quantification and statistical analysis".

**Acyl-biotin exchange (ABE).** Substitution of thioester-linked lipid modifications on cysteines with biotin was performed according to published protocols[33,42]. Briefly, Myc-tagged Rif1$_{NTD}$ was expressed from a *GAL1* promoter for 20 h by adding 2% (w/v) galactose to cell cultures in exponential growth phase. Cells from 100–200 ml cell culture were pelleted and lysed by bead-beating in lysis buffer (50 mM Tris-HCl pH 7.5, 150 mM KCl, 5 mM EDTA, 6 M urea) containing 50 mM *N*-ethylmaleimide (NEM, ThermoFisher Scientific, 23030) and protease inhibitors cocktail (Roche, 05 892 791 001). Triton X-100 was added to a final concentration of 0.5% to the cleared lysates, and free cysteines were blocked with NEM for 2 h at 4 °C on a rotating wheel. After chloroform-methanol precipitation, proteins were solubilized in resuspension buffer (50 mM Tris-HCl pH 7.5, 100 mM NaCl, 2% SDS, 8 M urea). Following addition of 1-biotinamido-4-[4'-(maleimidomethyl) cyclohexanecarboxamido]butane-biotin (BMCC-biotin, ThermoFisher Scientific, 21900) buffer (50 mM Tris-HCl pH 7.5, 300 μM BMCC-biotin, 1x PBS), samples were split and hydroxylamine (HA, Sigma-Aldrich, 438227) was added (+ HA) or not (-HA) to a final concentration of 1 M. Biotin exchange was performed for 2 h at 4 °C. After precipitation and solubilization, NeutrAvidin agarose beads (Thermo-Fisher Scientific, 29201) in 50 mM Tris-HCl pH 7.5, 5 mM EDTA, 0.1% Triton X-100 were added. Capture of biotinylated proteins (Avidin fraction, AviF) was performed overnight at 4 °C. Beads were washed with 1x PBS containing 0.5 M NaCl and 0.1% Triton X-100. For western blotting, resuspension buffer and 4x non-reducing sample buffer (240 mM Tris-HCl pH 8, 8% SDS, 40% glycerol, 0.02% bromophenol blue) were added and protein samples were boiled before SDS-PAGE. Aliquots of the +HA and -HA input fractions (prior to Avidin-capture) were used as loading controls and for normalization. See "Quantification and statistical analysis" for a detailed description of ABE quantification.

**Acyl-carbamidomethyl exchange (ACE).** Myc-tagged Rif1$_{NTD}$ expression and cell lysis was performed as described for ABE chemistry, with the exception that the lysis buffer contained 10 mM Tris(2-carboxyethyl)phosphine hydrochloride (TCEP, Fluka, 93284). TCEP was used to reduce disulfide bridges formed by closely-spaced cysteine residues, leaving *S*-acylation intact[66] (see also Supplementary Fig. 4). Final concentrations of 50 mM NEM and 0.5% Triton X-100 were added to the cleared lysates, and blocking of reactive cysteines was performed for 2 h at 4 °C on a rotating wheel. Chloroform-methanol precipitates were solubilized in resuspension buffer (50 mM Tris-HCl pH 7.5, 100 mM NaCl, 2% SDS, 8 M urea). Removal of *S*-acyl groups from cysteines was achieved by incubating the samples with beads buffer (50 mM Tris-HCl pH 7.5, 5 mM EDTA, 0.1% Triton X-100) containing 10 mM 1,4-dithiothreitol (DTT, Sigma-Aldrich, 43815) for 1 h at room temperature. Proteins were precipitated, resuspended, and beads buffer containing 20 mM 2-chloroacetamide (CAA, Sigma-Aldrich, 22790) was added, allowing CAM-labeling of freed-up cysteines. Myc-tagged Rif1$_{NTD}$ immunoprecipitation was performed overnight at 4 °C by addition of pre-washed anti-Myc-coupled magnetic agarose beads (Chromotek, ytma-20). After extensive washing with high (50 mM Tris-HCl pH 7.5, 500 mM NaCl) and low salt (50 mM Tris-HCl pH 7.5, 150 mM NaCl) buffers, samples were subjected to tryptic digestion on beads and analyzed by mass-spectrometry.

**ACE and acyl-*N*-ethylmaleimide exchange on synthetic peptides.** Synthetic peptides designed on the tryptic Rif1 peptide spanning residues 463–479 were obtained from ThermoFisher Scientific, and contained palmitoylated C466/CAM-labeled C473 (IYQC[PALMITOYL]IMLSPVC[CAM]ETIPEK), CAM-labeled C466/palmitoylated C473 (IYQC[CAM]IMLSPVC[PALMITOYL]ETIPEK) or palmitoylated C466/ palmitoylated C473 (IYQC[PALMITOYL]IMLSPVC[PAL-MITOYL]ETIPEK). Peptides were diluted in 20% CH$_3$CN, aliquoted, and stored at −20 °C. To monitor acyl-CAM and acyl-NEM exchange following treatment with reducing agents, peptide aliquots were diluted in a solution containing 50 mM Tris-HCl pH 8, 40% CH$_3$CN, and either 40 mM DTT or 40 mM TCEP, and reduction was carried out for 2 h at 56 °C. Alkylation with either 90 mM iodoacetamide (IAA, Sigma-Aldrich, I1149) or 90 mM NEM was performed for 1 h at room temperature. 1% trifluoroacetic acid (TFA, Pierce Perbio, 28904) was added, and samples were analyzed by mass spectrometry.

**Mass spectrometric analysis of synthetic peptides.** Synthetic peptides were analyzed by capillary liquid chromatography tandem mass spectrometry (LC-MS). Peptides were loaded in 0.1% formic acid (Pierce Perbio, 28905), 10% acetonitrile (VWR, 83640.290) in water (Sigma Aldrich, 14263–1 l) onto a 75 μm×15 cm ES811 column (Accucore C4, 2.6 μm, 150 Å) at a constant pressure of 800 bar, using an EASY-nLC 1000 liquid chromatograph with one-column set up (Thermo Scientific). Peptides were separated at a flow rate of 300 nl/min with a linear gradient of 10–60% buffer B in buffer A in 10 min, followed by a linear increase from 60 to 90% over 1 min, and the column was finally washed for 5 min at 90% buffer B (buffer A: 0.1% formic acid, 10% acetonitrile in water; buffer B: 0.1% formic acid in acetonitrile). The column was mounted on a DPV ion source (New Objective)

connected to an Orbitrap Fusion mass spectrometer (Thermo Scientific), data were acquired using 120,000 resolution, and MS1 signals were quantified using Skyline 4.1 (ref. [67]) to generate the results of Supplementary Fig. 4a.

**PRM data acquisition**. To increase sensitivity and specificity for the analysis of the biological material (Fig. 4), PRM analyses were performed using the LC-MS system described above, but with a 50 μm×15 cm ES801 column (C18, 2 μm, 100 Å) and a linear gradient of 2–6% buffer B in buffer A in 2 min, followed by an linear increase from 6 to 30% in 30 min, 30–50% in 10 min, 50–80% in 1 min, and finally the column was washed for 13 min at 80% buffer B at a flow rate of 150 nl/min. One MS spectrum at 120,000 resolution was acquired from 400–1200 Da, followed by 9 PRM spectra as described in Supplementary Table 1. An isolation window of 1.6 Da, a resolution of 120,000, and an automatic gain control value of 5e4 was used. Fragmentation was performed with a higher energy collision dissociation (HCD) collision energy of 30 eV, and MS/MS scans were acquired with a scan range of 100–2000 Da with a resolution of 120,000. Five non-modified peptides were selected from a data-dependent analysis using MASCOT for identification and Scaffold for validation (see below). These peptides were used as loading controls and for normalization of the three modified peptides, and are listed in Supplementary Table 1.

**PRM data analysis**. PRM data were processed using Skyline 4.1 (ref. [67]). The transition selection was systematically verified and adjusted when necessary to ensure that no co-eluting contaminant distorted quantification, based on traces co-elution (retention time), and the correlation between the relative intensities of the endogenous fragment ion traces and their counterparts from the library. Further calculations and figures were based on total integrated PRM intensities of the selected transitions (see Supplementary Fig. 4b, c). Mascot v. 2.5 (Matrix Science Ltd.) was used in the Decoy mode to search the Swissprot yeast version 2017_04 including common contaminants. The enzyme specificity was set to trypsin, allowing for up to one incomplete cleavage site. Modification of cysteines with carbamidomethyl (CAM; + 57.0245 Da), N-ethylmaleimide (NEM; +125.0477 Da), oxidation of methionine (+15.9949 Da), and acetylation of the protein N-terminus (+42.0106 Da) were set as variable modifications. Parent ion mass tolerance was set to 5 ppm and fragment ion mass tolerance to 0.01 Da. The results were validated with the program Scaffold Version 4.6.2 (Proteome Software, Portland, USA). Peptide identification was accepted if established at less than 0.1% false discovery rate, as calculated in Scaffold.

**Subcellular fractionation**. Assessment of solubility of Myc-tagged $Rif1_{NTD}$ in wild-type or mutant strains was performed by differential centrifugation as reported[33]. Exponentially growing cells were collected, spheroblasted, and resuspended in 100 mM KCl, 50 mM HEPES-KOH pH 7.5, 2.5 mM $MgCl_2$, and 0.4 M sorbitol. After centrifugation (1500×g for 1 min at 4 ℃), pellets were resuspended in an equal volume of extraction buffer (100 mM KCl, 50 mM HEPES-KOH pH 7.5, 2.5 mM $MgCl_2$, 50 mM NaF, 5 mM $Na_4P_2O_7$, 0.1 mM $NaVO_3$, protease inhibitors). Spheroblasts were lysed on ice by addition of 0.25% Triton X-100, and the whole cell extract fraction was collected. Lysates were underlayered with half a volume of 30% sucrose, and centrifuged at 13,500×g for 10 min at 4 ℃. Pellets, corresponding to the membrane-bound fraction, and soluble fraction were separated. Equal amounts of protein were loaded on 3–7% Tris-acetate gels. See "Quantification and statistical analysis" for a detailed description of the quantification of $Rif1_{NTD}$ enrichment in the soluble and membrane-bound fraction.

**Spinning-disk confocal microscopy**. Exponentially growing cells, cultured in sterile-filtered YPAD, were treated with 100 μg/ml Zeocin for 30 min or 100 Gy of IR in a Cellrad X-ray irradiator (Faxitron). Aliquots were taken before treatment and at defined time-points after treatment, and cells were collected by centrifugation and fixed with 4% paraformaldehyde at room temperature. Nuclear staining was done with 100 ng/ml DAPI (Sigma-Aldrich, D9542) in 1x PBS. To assess protein localization to the inner nuclear membrane, $\theta$ nuclei that arise from Nup53 overexpression were scored as described previously[44]. Briefly, exponentially growing cells in sterile-filtered YPLG were induced to overexpress Nup53 and/or Pfa4-GFP by addition of 2% (w/v) galactose for 20 h and $\theta$ nuclei were quantified and expressed relative to a Sec61-GFP positive control. Images were acquired using an Olympus IX81 spinning disk fluorescence confocal microscope equipped with a Yokogawa CSU-X1 scan head, a 2X back-illuminated EM-CCD EvolveDelta camera (Photometrics), an XY-motorized ASI MS-2000 Z-piezo stage, and a PlanApo x100, NA 1.45 oil objective. Fluorophores were excited at 561 nm (RFP and Ruby2), 491 nm (GFP), 445 nm (CFP), and 405 nm (DAPI). Z-stacks were collected with 100–300 ms exposures, with 40 slices at 0.25 μm intervals. Images were collected using Visiview v. 3.0 (Vistron systems GmbH), deconvolved using Huygens Remote Manager where necessary, and channel-aligned using Huygens Professional. Image analysis was performed using Fiji software[68]. Zoning assays were performed as described previously[45] using the Pointpicker ImageJ plugin, binning cells with single $Rif1_{NTD}$-GFP foci localized on the focal plane containing the largest nuclear outline marked by Nup49-Ruby2.

**Quantification and statistical analysis**. Statistical analyses were performed using GraphPad Prism v. 7. The applied tests and number of independent observations

are indicated in the corresponding figure legends. Western blot band intensities were quantified using Fiji software[68], and normalized using the indicated loading controls. Results shown in Supplementary Figs. 1a, 2c, 3b, 6e, and 6g were plotted with GraphPad Prism v. 7. Western blot signals for biotin-captured $Rif1_{NTD}$ (AviF) were quantified using Fiji software[68]. Band intensities were background-corrected by subtraction of the signal obtained in the AviF without HA and normalized to the input for data shown in Fig. 4b. Western blot signals for Rif1-enrichment in soluble or membrane fractions were quantified using Fiji software[68]. Band intensities were background-corrected and normalized to tubulin for the soluble fraction and to the nuclear pore p90 band for the membrane-bound fraction. Results are shown in Supplementary Fig. 5. Scanned EMSA phosphorimages were analyzed with ImageJ[69]. Total intensity of each individual lane was plotted and separated into the unbound DNA signal and the retarded DNA signal. The percentage of retarded DNA signal including all shifted bands (defined as "fraction bound") was analyzed and plotted using GraphPad Prism v. 7 to produce the results shown in Fig. 3d. For quantitation of $\theta$ nuclei and $Rif1_{NTD}$-GFP foci presented in Figs. 5 and 6, respectively, confocal microscopy images were analyzed using Fiji software[68]. Violin plots shown in Fig. 6c were generated with Rstudio.

**Reporting summary**. Further information on research design is available in the Nature Research Reporting Summary linked to this article.

## Data availability

A reporting summary for this article is available as a Supplementary Information file. The data underlying Figs. 1b–d, 2c, 3a, b, d, 4d, e, and 6b, c are provided as a Source Data file. All data supporting the findings of this study are available from the corresponding author upon reasonable request. Structural figures were prepared with PyMOL v. 1.8.4 (Schrödinger Inc.) and are based on published structures (PDB: 5NVR and PDB: 5NW5)[3]. The mass spectrometry proteomics data have been deposited at ProteomeXchange via PRIDE[70] and can be accessed using access code PXD012137.

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

## Acknowledgements

We thank K. Shimada, C. Horigome, A. Cheblal, and A. Seeber for technical help, and S. Gasser for generously sharing yeast strains and reagents. We are grateful for the assistance of J. Seebacher and V. Iesmantavicius from the Proteomics and Protein Analysis platform, and of L. Gelman, S. Bourke, and J. Eglinger from the Facility for Advanced Imaging and Microscopy at the Friedrich Miescher Institute. J.K.R. was supported by a Boehringer Ingelheim Fonds PhD fellowship. This work received funding to N.H.T. from the European Research Council (ERC) under the European Union's Horizon 2020 research and innovation program (grant agreement no. 666068), and to N.H.T. and D.S. from the Swiss National Science Foundation (Sinergia grant CRSII3_160734). Work in the laboratories of N.H.T. and U.R. was supported by the Novartis Research Foundation.

## Author contributions

G.A.F. conducted most experiments, with assistance by D.K.; G.A.F. established ACE with the help of D.H., who performed mass spectrometry. D.S and S.M. provided ChIP and replication control data for Rif1. N.H.T., J.K.R., and D.K. purified Rif1 proteins and carried out DNA-binding experiments. B.F. helped with data analyses. G.A.F. and U.R. conceived the study and wrote the manuscript with input from all authors.

## Additional information

**Competing interests:** The authors declare no competing interests.

