## [Peer Review File · Nature Communications]

Reviewers' comments:

Reviewer #1 (Remarks to the Author):

Fontana et al. seek to understand the effect of a post-translational modification, S-acylation, on the subcellular compartmentalization of DSB repair by NHEJ involving Rif1 (Rap1-interacting factor 1). Budding yeast Rif1 plays diverse roles in telomere homeostasis, DNA replication, and DSB repair. A previous study (Park et al. 2011) had reported that Pfa4 palmitoylates Rif1 and affects its localization to the nuclear periphery, and recently this group (Mattarocci et al, 2017) demonstrated that the Rif1 N-terminal domain (NTD) was required for NHEJ in yeast, presenting an elegant “shepherd’s crook” structure of the Rif1-NTD.

This paper identifies two Pfa4-dependent S-acylation sites in the Rif1-NTD, C466 and C473, and demonstrates that these sites are required for Rif1 function in NHEJ. Mutation of both cysteines to alanine resulted in decreased Rif1 accumulation at DSBs, increased ssDNA at an induced DSB, and decreased NHEJ, while dsDNA binding in vitro was unaffected. Acyl exchange chemistry was used to show that ~5-10% of Rif1 is palmitoylated on one (but not both) cysteines, suggesting redundancy in these residues’ role in NHEJ.

The experiments are convincing and for the most part well-controlled, the mapping of the acylation sites and their requirement for Rif1 function in NHEJ is solid, and this work will be of some interest within the DSB repair pathway choice and subcellular compartmentalization fields. However, given the previous work mentioned above, this paper represents an incremental increase in our understanding of Rif1 regulation and function. Additionally, there are several gaps in the experimentation (discussed below).

Major points

1. The authors state that Rif1-NTD is required and sufficient for promoting NHEJ, though sufficiency was not established in Mattarocci et al. (for example, by rescue of Rif1 Δ phenotypes with Rif1-NTD alone). Expression of Rif1-NTD and the mutants in a Rif1 Δ background could strengthen the paper by addressing the issue of Rif1-NTD sufficiency. Additionally, the labelling of Figure 3a is quite confusing. At present, it looks like the red bar represents Rif1-NTD being put back into a Rif1 Δ , which would suppress ssDNA formation. I’m sure this is not the intent of the authors, and that this bar actually represents a Rif1 Δ alone.
2. In Figure 3b, how is it that Rif1 occupancy reduced by 40%, as stated in the text? It looks more like a 4% reduction at the 2 h time point 0.3 kb from the cut. The authors observe a similar reduction in the 3K DNA-binding-defective “HOOK” mutant. Why is the reduction in Rif1 ChIP not greater? Do the acylation mutant and the DNA-binding mutant reduce ChIP signal for different reasons (ie that the acylation mutant fails to anchor at the nuclear periphery and that the DNA-binding mutant fails to engage DNA ends)? The differences between these mutants could be explored more, perhaps in localization studies (see point 4).
3. Figure 4 is compelling in showing that Pfa4-dependent acylation of Rif1 occurs in vivo, but given that the C466A/C473A effect is weaker than the Pfa4 Δ (Fig 4b), the authors could augment the epistasis analysis—perhaps by pairing the C466A/C473A with Pfa4 Δ —and distinguish the acylation pathway from the DNA-binding pathway.
4. The data on Rif1-GFP foci formation and localization to the nuclear periphery (Figure 5) is incomplete and only some of it is novel. It was previously shown that Rif1 focus formation at the nuclear periphery depends on Pfa4 (Park et al.), yet Figure 5c only quantifies Rif1-NTD-GFP localization in WT cells. It would be important to also confirm the previous result in Pfa4 Δ cells and extend it to the C466A/C473A mutant and the HOOK mutant. It might be predicted that C466A/C473A would phenocopy Pfa4 Δ , while the HOOK mutant wouldn’t change the subcellular localization, reflecting its defective DNA binding. Such a result would be crucial to support the claim that enrichment of Rif1 at the inner nuclear membrane is dependent on acylation by Pfa4 at these sites. Furthermore, it is overstated that Pfa4 Δ and the two mutants shown in 5b “strongly” compromise the ability of Rif1 to form foci. Many cells still have 1-4 foci, and no statistical

comparison is made between genotypes after DNA damage induction. As alluded to above, it would be crucial to know where these Rif1 foci localize within the zones described in 5c.

5. No statistical analyses are applied in Fig 1-4, and as stated in the previous point, the statistical analysis in Figure 5b does not address the comparisons that are made in the text—that is, between different genotypes.

Minor points

1. Intro first sentence: Rif1 (Rap1-interacting factor).
2. Concluding sentence of results section for Figure 1 should perhaps read “findings suggest that protein S-palmitoylation is important for NHEJ.” Pfa4 is shown to be required, but only through deletion of the gene, not disruption of S-palmitoylation specifically.
3. Figure 1b and/or 2b could include yku70 Δ (as in Mattarocci et al., as a negative control, showing the effect of total loss of NHEJ (much stronger than Rif1 effect)).
4. In Figure 3, the EMSAs could be explained better. How is the fraction bound quantified, given that there are two discrete bands falling within the bracket of “Rif1-bound”? Is the x-axis log scale (as stated in the legend), or not, as it is labelled? If not, is the 160 nM lane quantified and plotted or not? It should also be stated that the substrate is dsDNA, as this group has utilized dsDNA or dsDNA/ssDNA substrates in past EMSAs.
5. Acyl-CAM exchange (ACE) is utilized to good effect to show that one or the other (but not both) critical cysteines are acylated in vivo. The conclusion that these sites are likely redundant could be strengthened by performing ACE in the single mutants, where the remaining cysteine would be predicted to become more highly acylated. It might also be useful to address in the text the issue that the majority of cellular Rif1 appears not to be acylated at either site. These experiments appear to be done under undamaged conditions. Does S-acylation on Rif1 change after DNA damage is induced?
6. The many sample images in 5a seem unnecessary. Showing either zeocin or IR would be sufficient.
7. Pg. 10 Pfa4 Δ genotype is written “Pfa4 Δ ” with capital “A”.
8. Same paragraph, what is meant by 28% of cells display “mostly” a single focus?
9. Discussion: “DNA damage-induced Rif1 focus formation was dependent on Pfa4...” (missing “Rif1”).

Reviewer #2 (Remarks to the Author):

Protein S-acylation plays critical roles in regulating protein localization, activity, stability, and complex formation. Hence, it is involved in various biological processes such as signal transduction, apoptosis, and metabolism. In this study, Fontana et al. made an interesting and important discovery that the S-acylation of Rif1 plays a key role in regulating DNA double-strand break repair. Overall, the manuscript was beautifully written, the experiments were well designed, and the data are solid. However, several minor issues need to be addressed.

Page 6, Line 5. The conclusion that “protein S-palmitoylation is important for NHEJ efficiency in yeast” is an overstatement. Although Pfa4 is a palmitoyl acyltransferase, it might function in an S-palmitoylation-independent fashion. Moreover, protein S-palmitoylation was not investigated in the first section (i.e., The palmitoyl acyltransferase Pfa4 promotes NHEJ). Better to change the statement to “protein S-palmitoylation is potentially important for NHEJ efficiency in yeast”.

Page 6, Results section #2. The authors did a nice job in prioritizing a list of Cys residues as candidate S-palmitoylation sites. But was the evolutionary conservation of these Cys residues across different species taken into consideration? If the authors believe that the S-palmitoylation of Rif1 is important in DNA double-strand break repair, the S-palmitoylation sites should be largely conserved from yeast to human.

Pages 7 and 9. In page 7, the authors stated that “These results indicate an impairment of NHEJ by combined, but not individual, loss of potential Rif1 S-acylation sites C466 and C473.” However, in page 9, they concluded that “S-acylation occurs either at C466 or C473, but not – or very rarely – at both residues.” How can the findings be reconciled? The authors need to discuss this a bit.

Page 9, Line 17. Please replace “mass spectrometry” with “targeted mass spectrometry” or “parallel reaction monitoring”. In addition, the Methods sections for PRM development and application are not very clear. For example, how were the target peptides selected, and how were the PRM assays developed? In the “PRM data analysis” section, what was the purpose of performing Mascot database searching, and why was the palmitoyl (PALM; +238.2297) modification included in the list of the variable modifications? It seems that the authors either conducted a discovery proteomic analysis or attempted to quantify intact S-palmitoylated peptides, but probably had no success.

Page 26, Figure 3c. Why were the BSA bands present?

Page 32, Figure 6. What do the singly and dually S-palmitoylated forms (above the left “Peripheral DSB”) stand for? In addition, the finding that “S-acylation occurs either at C466 or C473, but not – or very rarely – at both residues” needs to be taken into consideration in the illustration.

Reviewer #3 (Remarks to the Author):

In this paper, Fontana et al. build on three previously reported observations: that Rif1 is involved in DNA double strand break repair, that Rif1 is acylated and that this modification is mediated by the palmitoyltransferase Pfa4. The paper aims at connecting these events. While the paper is very well written, the novelty, the mechanistic insight is somewhat limited, and therefore the study appears preliminary.

The authors show that Rif1 can be S-acylated on two specific cysteines in the NTD but probably others as well that are not identified. The palmitoylation evidence should be strengthened. The main conclusion that palmitoylated Rif1 mediates double strand break repair by accumulating at the inner nuclear membrane is insufficiently supported by the shown data. Given that the basis of the current findings had already been reported, more depth in mechanistic were expected.

- The model proposed by the authors (Fig 6) indicates that S-acylated Rif1 localizes at the inner nuclear membrane and interacts with dsDNA. Where does the acylation by Pfa4 occur? In the cytoplasm? And, thus, would palmitoylated Rif1 enter the nucleus? Or does Pfa4 localize to the inner nuclear membrane? So far, the presence of palmitoyltransferases in the nucleus has not been reported, and thus this would need to be documented by localization studies of Pfa4 by fluorescence or electron microscopy, by subcellular fractionation (membrane vs soluble) to see Rif1 distribution in the different backgrounds.
- ABE results show decreased but not completely abolished signal upon mutation of cysteines or Pfa4 deletion. Identification of all palmitoylation sites would be important. Also data that excludes the other palmitoyltransferases would strengthen the conclusion, excluding indirect effects of Pfa4. The ABE or derivatives thereof are not ideal to study proteins with multiple palmitoylation sites since the presence of a single remaining site is in principle sufficient to bring down the protein. So, alternative methods should be used to confirm these findings.
- Mass spectrometry was used to analyze palmitoylation. While this technique is elegant, the conclusion that the two sites cannot be simultaneously modified would need confirmation by other methods such as PEGylation (a method established by various groups including that of Fukata and that of Huang), which allows an estimation of the site occupancy and also what percentage of Rif1 is actually modified in the cell.
- As mentioned by the authors in the manuscript, palmitoylation is reversible and can be dynamic.

The dynamic aspect is missing in the current work. The key question is raised by the authors in their conclusion; what is the “dynamic regulation of Rif1 by Pfa4 in response to DNA damage”. Is the acylation status of Rif1 modified upon induction of DNA damage? This should be address in detail, looking at the different palmitoylation sites. The mass spec and the ACE analyses were done it seems on untreated cells.

- Statistical analysis is missing in figures 1 to 4. T-test-associated p-values should be provided for all discussed comparisons. Also, statistical analysis shown in Figure 5 seems inconsistent with the conclusions drawn from the data. The authors should provide the p-values associated to the comparisons discussed in the text. Additionally, the 300 cells obtained from 3 experiments cannot be processed in the same way for statistical purposes as they are not truly independent.
- Fig 5c: the distribution of Rif1 signal between the three defined zones should be compared to the non-treated condition. How would this distribution look in the Pfa4-deletion or the Rif1 mutant backgrounds?

Minor comments:

- It is not clear why the authors resort to DTT to cleave the thioester bonds in their ACE method instead of using hydroxylamine as for ABE. Also, the MS data provided in Fig S4 supporting TCEP does not cleave thioester bonds is not convincing.
- It would be nice to see an example of the raw data used for the plots provided in Figs 1, 3, and S4.
- Catalog numbers (or even better RRID) should be provided for all antibodies used, and ideally other reagents (as those used in ABE/ACE methods).
- Using the label “RIF1” to refer to the WT strain in Fig 1 is confusing. Similarly, labels “Rif1NTD in delta-Rif1” in Fig 3a is misleading.

We would like to thank all reviewers for constructive feedback on our manuscript “*Rif1 S-acylation mediates DNA double-strand break repair at the inner nuclear membrane*”. We appreciate the comments on the importance of our work and have addressed the points raised in our experimental revision as detailed below. (Textual changes in the revised MS are highlighted yellow.)

Reviewers' comments:

Reviewer #1 (Remarks to the Author):

Fontana et al. seek to understand the effect of a post-translational modification, S-acylation, on the subcellular compartmentalization of DSB repair by NHEJ involving Rif1 (Rap1-interacting factor 1). Budding yeast Rif1 plays diverse roles in telomere homeostasis, DNA replication, and DSB repair. A previous study (Park et al. 2011) had reported that Pfa4 palmitoylates Rif1 and affects its localization to the nuclear periphery, and recently this group (Mattarocci et al, 2017) demonstrated that the Rif1 N-terminal domain (NTD) was required for NHEJ in yeast, presenting an elegant “shepherd’s crook” structure of the Rif1-NTD.

This paper identifies two Pfa4-dependent S-acylation sites in the Rif1-NTD, C466 and C473, and demonstrates that these sites are required for Rif1 function in NHEJ. Mutation of both cysteines to alanine resulted in decreased Rif1 accumulation at DSBs, increased ssDNA at an induced DSB, and decreased NHEJ, while dsDNA binding in vitro was unaffected. Acyl exchange chemistry was used to show that ~5-10% of Rif1 is palmitoylated on one (but not both) cysteines, suggesting redundancy in these residues’ role in NHEJ.

The experiments are convincing and for the most part well-controlled, the mapping of the acylation sites and their requirement for Rif1 function in NHEJ is solid, and this work will be of some interest within the DSB repair pathway choice and subcellular compartmentalization fields. However, given the previous work mentioned above, this paper represents an incremental increase in our understanding of Rif1 regulation and function. Additionally, there are several gaps in the experimentation (discussed below).

Authors’ response: We thank reviewer 1 for recognizing our work on identifying Rif1 S-acylation as a novel mechanism for spatial control of chromosome break repair will be of interest to both the DSB repair pathway choice and subcellular compartmentalization fields. In response to the comments provided, we have revised the manuscript as detailed point-by-point below.

Major points

1. The authors state that Rif1-NTD is required and sufficient for promoting NHEJ, though sufficiency was not established in Mattarocci et al. (for example, by rescue of Rif1 Δ phenotypes with Rif1-NTD alone). Expression of Rif1-NTD and the mutants in a Rif1 Δ background could strengthen the paper by addressing the issue of Rif1-NTD sufficiency. Additionally, the labelling of Figure 3a is quite confusing. At present, it looks like the red bar represents Rif1-NTD being put back into a Rif1 Δ , which would suppress ssDNA formation. I’m sure this is not the intent of the authors, and that this bar actually represents a Rif1 Δ alone.

Authors’ response: Rif1-NTD sufficiency in promoting NHEJ is the basis for our genetic/functional deduction of NHEJ-relevant S-acylation sites in Rif1, and we thank reviewer 1 for pointing out that Rif1-NTD NHEJ sufficiency has not been made adequately clear in the current MS. In a previous study (Mattarocci et al. (2017) Nat Struct Mol Biol 24:588-595), we have shown that cells expressing Rif1-NTD exhibit NHEJ efficiencies indistinguishable from cells expressing the full-length wild-type protein

(both from the endogenous *RIF1* locus), demonstrating that the NTD is sufficient to support the role of Rif1 in promoting NHEJ. This information is contained in the Supplementary Information of the 2017 study (Supplementary Figure 5b), not in the main text, potentially lacking visibility. Rif1-NTD NHEJ-sufficiency is consistent with Rif1-NTD being able to attenuate DNA end-resection at DSBs in the present study, an ability that is lost upon deleting *PFA4* or endogenously introducing the S-acylation or DNA-binding site mutations into the *RIF1* locus expressing Rif1-NTD (**Fig. 3a**). The experiments in Mattarocci et al. (2017) Nat Struct Mol Biol 24:588-595, and the current work assess the ability of Rif1 variants to promote NHEJ in absence of wild-type Rif1. In each case, Rif1-NTD – or mutants thereof – are the only version of Rif1 cells express, showing that Rif1-NTD but not Rif1-NTD containing the C466A/C473A or K437E/K563E/K570E mutations rescue *rif1Δ*-associated NHEJ phenotypes. In the revised MS, we have further clarified the issue by explicitly referring to observations in Mattarocci et al. (2017) Nat Struct Mol Biol 24:588-595 that establish sufficiency of Rif1-NTD for Rif1-mediated NHEJ efficiency. In addition, we have included in the revised MS a side-by-side comparison of NHEJ efficiency in cells expressing genomic versions of full-length Rif1 or Rif1-NTD only, for additional clarity on Rif1-NTD NHEJ sufficiency. This data has been included in the revised MS in **new Supplementary Fig. 2a**. The revised text on page 6 now reads:

*“This analysis was based on two assumptions: first, relevant S-acylated cysteine residues must be contained within the N-terminal domain of Rif1 (residues 1-1322, hereafter referred to as Rif1_{NTD} for Rif1 N-terminal domain). This region of Rif1 was shown to be required and sufficient for promoting NHEJ, with cells expressing Rif1_{NTD} from the endogenous RIF1 locus being as effective in promoting NHEJ as cells expressing full-length Rif1⁸ (see also **Supplementary Fig. 2a**).”*

With regard to **Figure 3a**, we agree the labeling was unclear. Indeed, the red bar represents *rif1Δ*, not *rif1Δ* expressing Rif1-NTD. The annotation of **Fig. 3a** has now been fixed to remove this ambiguity, and **Fig. 3a** legend was amended to read:

“DNA end-resection upon DSB induction at the MAT locus in strains expressing wild-type (WT) Rif1_{NTD}, the indicated Rif1_{NTD} mutants, or deleted for RIF1.”

2. In Figure 3b, how is it that Rif1 occupancy reduced by 40%, as stated in the text? It looks more like a 4% reduction at the 2 h time point 0.3 kb from the cut. The authors observe a similar reduction in the 3K DNA-binding-defective “HOOK” mutant. Why is the reduction in Rif1 ChIP not greater? Do the acylation mutant and the DNA-binding mutant reduce ChIP signal for different reasons (ie that the acylation mutant fails to anchor at the nuclear periphery and that the DNA-binding mutant fails to engage DNA ends)? The differences between these mutants could be explored more, perhaps in localization studies (see point 4).

Authors’ response: We thank reviewer 1 for spotting the mistake on page 8 referring to the ChIP experiment shown in **Figure 3b**. We are indeed referring not to the 2 h but the 4 h time-point after cut, and a 40% reduction in Rif1 occupancy at DSBs relative to wild-type. We have revised the text on page 8 accordingly:

*“As expected, Rif1_{NTD} accumulated following cut induction, but occupancy compared to wild-type was reduced by ~40% (4 h time-point, 0.3 kb from DSB) when potential Rif1 S-acylation sites C466 and C473 were mutated, or when PFA4 was deleted (**Fig. 3b**).”*

The Rif1 HOOK mutant is a hypomorphic DNA binder with residual *in vitro* DNA-binding activity (see **Fig. 3d** herein and Fig. 2 in Mattarocci et al. (2017) Nat Struct Mol Biol 24:588-595). Yet, its remaining DNA-binding activity and DSB recruitment *in vivo* is not sufficient to support Rif1’s role in attenuating DNA end-resection and promoting NHEJ (**Fig. 3** herein and Mattarocci et al. (2017) Nat

Struct Mol Biol 24:588-595, Fig. 4). In light of this, a NHEJ null-phenotype is not necessarily accompanied by complete loss of Rif1-DSB recruitment as assessed by ChIP. Indeed, we now observe that Rif1 S-acylation mutant C466A/C473A – similar to the Rif1 HOOK mutant – exhibits compromised and not abolished DSB occupancy (**Fig. 3b**), and this inability to engage with DSBs with wild-type characteristics is associated with complete loss of the ability to promote Rif1-mediated NHEJ (**Figs. 2b, 3a**). Yet, in contrast to the Rif1 HOOK mutant, the Rif1 S-acylation mutant is fully competent in DNA binding *in vitro* (**Fig. 3c, d**). We therefore agree with the interpretation that the Rif1 HOOK DNA-binding mutant and the Rif1 C466A/C473A S-acylation mutant fail to appropriately accumulate at, and productively engage with DSBs for different reasons; the latter due to impaired Rif1-inner nuclear membrane positioning. To address this directly, we have now created a new Rif1_{NTD} C466A C473A HOOK mutant (C466A C473A K437E K563E K570E, referred to Rif1_{NTD} S-acylation/HOOK mutant) and conducted ChIP experiments. We found that the Rif1_{NTD} S-acylation/HOOK mutant is consistently more compromised in DSB-recruitment than the Rif1_{NTD} C466A C473A S-acylation mutant. These new data corroborate the notion of separate contributions of Rif1 S-acylation and DNA binding to DSB-recruitment and have been **added as new Supplementary Figure 3d, e** (a combined Rif1_{NTD} HOOK/S-acylation mutant has also been used for subnuclear localization studies, see response to point 4 below). Textual changes to the MS on pages 8, 9 are as follows:

“As expected, Rif1_{NTD} accumulated following cut induction, but occupancy compared to wild-type was reduced by ~40% (4 h time-point, 0.3 kb from DSB) when potential Rif1 S-acylation sites C466 and C473 were mutated, or when PFA4 was deleted (Fig. 3b). Consistent with previous results⁸, a similar loss of Rif1_{NTD} from the DSB was observed upon introduction of the DNA-binding site HOOK mutation (Fig. 3b), and combining the HOOK and C466A/C473A mutants reduced Rif1_{NTD} occupancy at DSBs further (Supplementary Fig. 3d, e).”

“These results show that the potential Rif1 acceptor site C466/C473 for Pfa4-dependent S-acylation and the HOOK domain’s DNA-binding site make separate contributions to Rif1’s ability to effectively engage an induced DSB, while the integrity of both sites is indispensable for Rif1-mediated DNA end-protection and NHEJ.”

3. Figure 4 is compelling in showing that Pfa4-dependent acylation of Rif1 occurs *in vivo*, but given that the C466A/C473A effect is weaker than the Pfa4Δ (Fig 4b), the authors could augment the epistasis analysis— perhaps by pairing the C466A/C473A with Pfa4Δ—and distinguish the acylation pathway from the DNA-binding pathway.

Authors’ response: We thank reviewer 1 for commenting on the compelling evidence of Rif1 palmitoylation at C466 and C473 *in vivo* presented in **Fig. 4**; these sites represent the first *bona fide* S-acylation sites mapped within Rif1. C466 and C473 were identified by mutational analysis as essential for NHEJ (**Figs. 2 and 3**), and we used acyl-biotin exchange (ABE) followed by Western blotting, and then acyl-chloracetamide exchange (ACE) followed by targeted mass spectrometry to prove *in vivo* S-acylation at these residues. While ABE assays indicate that sites other than those relevant for NHEJ, i.e. C466 and C473, may be S-acylated in a Pfa4-dependent manner, having mutated and tested all potential Rif1_{NTD} palmitoylation sites in NHEJ assays, we can conclude these potential additional sites do not impact Rif1-promoted NHEJ if C466/C473 are intact, nor do they support NHEJ if C466/C473 are mutated. This was not sufficiently clear, and we now point this out in the revised text as follows on page 14:

*“We demonstrate that C466 and C473 are modified by S-acylation *in vivo* in a strictly Pfa4-dependent manner, strongly suggesting they are direct Pfa4 S-palmitoylation targets (Fig. 4). Residual avidin-enrichment after subjecting Rif1_{NTD} C466A/C473A to ABE chemistry points to Rif1 S-acylation at*

additional, as-yet unmapped sites (Fig. 4b). While these potential additional sites are neither required nor sufficient for Rif1 to promote NHEJ, at least one Pfa4-dependent S-acylation event at C466 or C473 is essential for Rif1-mediated NHEJ (Fig. 2)."

Pairing the C466A/C473A mutation with a deletion of *PFA4* could provide a pathway of assessing whether S-acylation at C466 and C473 is dependent upon Pfa4 by ABE chemistry. However, we provide the more direct demonstration of Pfa4-dependent S-acylation at these residues by our newly devised ACE method using targeted mass spectrometry. In **Fig. 4d**, we show that mass spectrometry signals identifying C466 and C473 S-acylation are lost upon deletion of *PFA4*. In line with this are our new data showing that *PFA4* is the only palmitoyl transferase out of seven DHHC family enzymes present in yeast whose deletion results in Zeocin resistance, an indicator of compromised NHEJ (see **new Supplementary Fig. 1b**).

Regarding a distinction between the involvement of DNA-binding by the Rif1 HOOK domain and Rif1 S-acylation at C466 and C473 in Rif1's response to DSBs, we have now further interrogated this in our revisions by generating the combined Rif1_{NTD} S-acylation/HOOK mutant (Rif1 C466A C473A K437E K563E K570E). While individually, the C466A/C473A and HOOK mutations result in a NHEJ null-phenotype, we could reveal that the Rif1 HOOK DNA-binding domain and Rif1 S-acylation make separate contributions to DSB recruitment (**new Supplementary Figure 3d, e**, see also point 2 above).

4. The data on Rif1-GFP foci formation and localization to the nuclear periphery (Figure 5) is incomplete and only some of it is novel. It was previously shown that Rif1 focus formation at the nuclear periphery depends on Pfa4 (Park et al.), yet Figure 5c only quantifies Rif1-NTD-GFP localization in WT cells. It would be important to also confirm the previous result in Pfa4Δ cells and extend it to the C466A/C473A mutant and the HOOK mutant. It might be predicted that C466A/C473A would phenocopy Pfa4Δ, while the HOOK mutant wouldn't change the subcellular localization, reflecting its defective DNA binding. Such a result would be crucial to support the claim that enrichment of Rif1 at the inner nuclear membrane is dependent on acylation by Pfa4 at these sites. Furthermore, it is overstated that Pfa4Δ and the two mutants shown in 5b "strongly" compromise the ability of Rif1 to form foci. Many cells still have 1-4 foci, and no statistical comparison is made between genotypes after DNA damage induction. As alluded to above, it would be crucial to know where these Rif1 foci localize within the zones described in 5c.

Authors' response: Following this reviewer's and reviewer 3's suggestion, we now present a direct comparison of the fraction of Rif1 focus-positive cells in wild-type cells, and cells expressing the Rif1 C466A C473A S-acylation mutant, the Rif1 HOOK DNA-binding mutant, and also the new Rif1 S-acylation/HOOK C466A C473A K437E K563E K570E mutant before and after DNA damage treatment (**new panels b and c of revised Figure 6**), with added statistical analysis for inter-strain comparisons. We thank the reviewers for this suggestion, which shows in much clearer fashion how significantly the DNA damage-induced component of Rif1 focus formation is diminished in all tested mutants compared to wild-type. The text has been revised accordingly (starting at the bottom of page 11):

*"After DNA-damage treatment with Zeocin or ionizing radiation (IR), focus formation was strongly induced, reaching a peak ~30 min post-treatment (**Supplementary Fig. 6b**), when ~60% (Zeocin) and ~80% (IR) of cells exhibited Rif1_{NTD}-GFP foci (**Fig. 6a, b and Supplementary Fig. 6c, d**). Moreover, while the majority of focus-positive cells in unperturbed conditions contained a single Rif1_{NTD}-GFP focus, most focus-positive cells exhibited multiple (up to four) foci after DNA damage treatment (**Fig. 6c**). DNA damage-induced Rif1_{NTD}-GFP foci were observed in G1 and S/G2 cells with no overt cell-cycle dependence. DNA-damage treatment did not lead to increased Rif1_{NTD} expression levels (**Supplementary Fig. 6e**), suggesting that focus formation reflected the redistribution of Rif1_{NTD}-GFP*

into foci upon DNA damage. Next, we analyzed *Rif1*_{NTD}-GFP foci in *pfa4Δ* cells. Untreated cells were indistinguishable from wild-type with ~30% *Rif1*_{NTD}-GFP focus-positive cells, the majority of which contained a single *Rif1*_{NTD}-GFP focus (**Fig. 6b, c** and **Supplementary Fig. 6c**). In contrast, the ability to form DNA damage-induced *Rif1*_{NTD}-GFP foci (~36% and ~38% focus-positive cells after Zeocin and IR, respectively, see **Fig. 6b** and **Supplementary Fig. 6d, f**) and the formation of multiple *Rif1* foci in response to DNA damage (**Fig. 6c**) was significantly abrogated in *pfa4Δ* cells. Protein levels of *Rif1*_{NTD} remained unchanged upon loss of Pfa4 (**Supplementary Fig. 6g**). Importantly, in Pfa4-proficient cells, introducing the S-acylation mutation C466A/C473A also diminished the formation of DNA damage-induced *Rif1*_{NTD} foci and the ability of cells to form multiple *Rif1* foci (**Fig. 6b, c**). Like S-acylation mutant *Rif1*_{NTD} C466A/C473A, the *Rif1* HOOK DNA-binding mutant was strongly compromised in its ability to form foci in response to DNA-damage treatment (**Fig. 6b, c**). Combining the S-acylation and HOOK mutations led to a more severe phenotype compared to either the *Rif1*_{NTD} C466A/C473A or the *Rif1* HOOK mutant (**Fig. 6b, c**). Thus, Pfa4-dependent S-acylation of *Rif1* at C466/C473 and the ability of *Rif1* to bind DNA contribute to effective *Rif1* accumulation upon DNA damage.”

It is due to this significant defect in the DNA damage-induced component of *Rif1* focus-formation in all mutants analyzed that we initially concentrated our description of the subnuclear localization of this completely novel type of *Rif1* foci (independent of previously described *Rif1* signals at telomeres) on wild-type cells. Here, DNA damage-induced foci are prominent and show a striking bias for peripheral localization (zone 1) in zoning assays. Residual *Rif1* foci in all mutants analyzed are mostly those that are observed spontaneously in all strains and, by definition, not the DNA-damage induced component of *Rif1* foci that we have discovered and describe in the current study. Nonetheless, there appears to be some residual *Rif1* focus-formation induced upon DNA damage also in the mutants, and we have now extended our nuclear zoning as suggested. We now present analyses of the wild-type as well as *Rif1*_{NTD} in a *pfa4Δ* background, the *Rif1*_{NTD} C466A C473A S-acylation mutant, the *Rif1*_{NTD} HOOK DNA-binding mutant, and the newly created combined *Rif1*_{NTD} S-acylation/HOOK mutant (*Rif1* C466A C473A K437E K563E K570E), evaluating in each case where *Rif1*_{NTD} foci localize. As shown previously, in wild-type cells, the majority of *Rif1* foci is found in zone 1, i.e. immediately adjacent to the inner nuclear membrane. In contrast, all S-acylation-compromised mutants (including *pfa4Δ* and cells expressing *Rif1*_{NTD} C466A C473A and *Rif1*_{NTD} C466A C473A K437E K563E K570E) exhibited a greater proportion of foci situated in zone 2 (more distant to the inner nuclear membrane). Interestingly, the *Rif1* DNA-binding HOOK mutant did not show this trend of redistribution from zone 1 to zone 2 of residual foci, confirming the prediction of reviewer 1. These new data (presented in **new Figure 6, panel d**) are consistent with Pfa4-dependent S-acylation at C466/C473 promoting attachment at the inner nuclear membrane and with S-acylation-mediated *Rif1* sequestration at the inner nuclear membrane being maintained in DNA-binding mutants.

The text (page 13) has been revised to read:

“To determine the sub-nuclear localization of *Rif1*_{NTD}-GFP foci, we scored their position relative to the nuclear envelope marked by fluorescently tagged nuclear-pore component *Nup49*⁵⁹. Dividing the nucleus into three concentric zones of equal area, we found a strong bias of *Rif1* accumulation in outermost zone 1, at the nuclear periphery, in wild-type cells. Upon DNA damage, ~60% of *Rif1*_{NTD}-GFP foci localized in zone 1 (**Fig. 6d**). While cells expressing the *Rif1* HOOK DNA-binding mutant maintained a strong localization bias to zone 1, cells with compromised *Rif1* S-acylation expressing *Rif1*_{NTD} C466A/C473A, *Rif1*_{NTD} C466A/C473A HOOK, or wild-type *Rif1*_{NTD} in a *pfa4Δ* background, exhibited an increase in zone 2-localized foci at the expense of zone 1-localized foci. Thus, *Rif1* S-acylation mutants display an apparent reduction in *Rif1*-inner nuclear membrane interactions in conjunction with a significant impairment in the formation of DNA damage-induced *Rif1*-foci observed in all mutant backgrounds tested (**Fig. 6b, c** and **Supplementary Fig. 6c, d, and f**). Taken

together, these data are consistent with a model where enrichment of Rif1 at the inner nuclear membrane mediated by Pfa4-dependent S-acylation of C466/C473 and its intrinsic DNA-binding activity enable effective Rif1 accumulation at nuclear-peripheral DNA damage, promoting preferential repair of membrane-proximal DSBs along the NHEJ pathway (Fig. 7)."

Further support for enrichment of Rif1 at the inner nuclear membrane being dependent on S-acylation by Pfa4 at Rif1 C466 and C473 comes from additional new data showing by subcellular fractionation that – compared to wild-type – Rif1 is increased in the soluble fraction and reduced in the membrane fraction upon loss of Pfa4, with Rif1 C466A C473A exhibiting an intermediate phenotype (see **new Supplementary Fig. 5**). This is referred to in the revised MS on page 11:

"Using cell fractionation, we confirmed previous results⁴² showing Pfa4-dependent membrane associations of Rif1, which proved partially dependent on Rif1 residues C466 and C473 (Supplementary Fig. 5)."

5. No statistical analyses are applied in Fig 1-4, and as stated in the previous point, the statistical analysis in Figure 5b does not address the comparisons that are made in the text—that is, between different genotypes.

Authors' response: In our revisions, we have increased the number of biological replicates and included appropriate statistical tests for all assays shown in **Fig. 1, panels b-d, Fig. 2c, Fig. 3a-b, Fig. 6b-d** as detailed in the respective figure legends in our revised MS. The statistical analysis for comparisons between strains for Rif1 focus formation upon DNA damage has been added as suggested and is now presented in **revised Figure 6, new panel b**.

Minor points

1. Intro first sentence: Rif1 (Rap1-interacting factor).

Authors' response: This has been corrected in the text on page 3.

2. Concluding sentence of results section for Figure 1 should perhaps read "findings suggest that protein S- palmitoylation is important for NHEJ." Pfa4 is shown to be required, but only through deletion of the gene, not disruption of S-palmitoylation specifically.

Authors' response: This change has been included in the revised text on page 6.

3. Figure 1b and/or 2b could include yku70Δ (as in Mattarocci et al., as a negative control, showing the effect of total loss of NHEJ (much stronger than Rif1 effect)).

Authors' response: The yku70Δ control has now been included in **revised Fig. 1b** and is referred to in the text on page 5:

"As expected, cell viability upon DSB induction was fully dependent on core NHEJ factor Ku70 (Fig. 1b)."

4. In Figure 3, the EMSAs could be explained better. How is the fraction bound quantified, given that there are two discrete bands falling within the bracket of "Rif1-bound"? Is the x-axis log scale (as stated in the legend), or not, as it is labelled? If not, is the 160 nM lane quantified and plotted or

not? It should also be stated that the substrate is dsDNA, as this group has utilized dsDNA or dsDNA/ssDNA substrates in past EMSAs.

Authors' response: The fraction Rif-bound DNA includes all gel-shifted DNA species. An extended description of EMSA quantification has been included in the revised MS, **Online Methods**, on page 12:

*“Scanned EMSA phosphorimages were analyzed with ImageJ⁹¹. Total intensity of each individual lane was plotted and separated into the unbound DNA signal and the retarded DNA signal. The percentage of retarded DNA signal including all shifted bands (defined as “fraction bound”) was analyzed and plotted using GraphPad Prism v. 7 to produce the results shown in **Fig. 3d**.”*

The X-axis for the graph showing EMSA quantifications in **Figure 3d** is log₂ scale (160 nM protein concentration is omitted from the quantitation because no unbound DNA remains for the wild-type). The DNA substrate used (dsDNA) has now been unambiguously identified in the legend of **Figure 3d**.

5. Acyl-CAM exchange (ACE) is utilized to good effect to show that one or the other (but not both) critical cysteines are acylated in vivo. The conclusion that these sites are likely redundant could be strengthened by performing ACE in the single mutants, where the remaining cysteine would be predicted to become more highly acylated. It might also be useful to address in the text the issue that the majority of cellular Rif1 appears not to be acylated at either site. These experiments appear to be done under undamaged conditions. Does S-acylation on Rif1 change after DNA damage is induced?

Authors' response: A quantitative comparison between the tryptic peptides containing C466/C473 with or without mutation of one (C466A) or the other site (C473A) to alanine is complicated by the fact that these peptides have different sequences (WT: IYQCNEMIMLSPVCNEMETIPEK; C466A mutant: IYQANEMIMLSPVCNEMETIPEK; C473A mutant: IYQCNEMIMLSPVANEMETIPEK). Therefore, direct comparisons of S-acylation levels by our ACE method would not lead to an accurate estimation. However, our mutagenesis of Rif1 clearly shows that Rif1-mediated NHEJ is not impaired as long as either C466 or C473 are intact, yet fully disrupted when both sites are mutated together (**Fig. 2**), strongly indicating site-redundancy. To provide an estimate of overall palmitoylation levels, we have extended our analysis of the four different forms of the synthetic Rif1 peptide spanning residues 463-479 (unmodified, single modification at C466 or C473, double-modified). With MS2 spectra (in particular for fragments y3 and y9-y12) and retention times in HPLC being very similar for these peptides (see **new Supplementary Fig. 4b, c**), we integrated the signals for y3 and y9-y12 and further normalized data from biological samples with five non-modified Rif1 peptides (shown in **Supplementary Table 1** and PRM analysis shown in **Source data file**). Based on this analysis, we estimate that roughly 15-20% of cellular Rif1 is S-acylated at C466 or C473. In **revised Fig. 4d**, we present relative peptide signal (unmodified and modified) measured in wild-type vs. *pfa4Δ*, annotated with actual PRM readings (**new Fig. 4e**). As suggested, we have further extended our analysis by performing ACE experiments in parallel in untreated cells and cells treated with radiomimetic drug Zeocin. After DNA damage, our measurements indicate 15-20% of S-acylation of Rif1. These data are now included in the revised MS (**Fig. 4, new panel e**) and indicate that Rif1 is constitutively S-acylated, although measurements on the complete pool of cellular Rif1 may preclude detection of local S-acylation Rif1 dynamics. Our results have been uploaded to the PRIDE proteomics data repository (name: **PXD012137**) and can be accessed with username: reviewer56682@ebi.ac.uk; password: dbnUV59i. Textual changes to the MS as follows (page 10):

“The PRM values for double NEM-modified Rif1 peptides spanning residues 463 to 479 are approximately 10 times higher than the values for peptides derived from Rif1 S-acylated at C466 or

C473. This implies that a substantial fraction of 15-20% of Rif1 was S-acylated at either C466 or C473 in wild-type cells (**Fig. 4d**, see also **Supplementary Fig. 4**). DNA-damage treatment with Zeocin did not lead to gross changes in Rif1 S-acylation levels (**Fig. 4e**), consistent with constitutive Rif1 S-acylation.”

6. The many sample images in 5a seem unnecessary. Showing either zeocin or IR would be sufficient.

Authors' response: We have removed the examples showing IR-induced Rif1_{NTD}-GFP by confocal microscopy from **revised Fig. 6**, now only presenting Zeocin-induced foci in the main text. IR-induced foci are now included in **revised Supplementary Fig. 6, panel c**, quantification in **panel d**.

7. Pg. 10 Pfa4Δ genotype is written “PfA4Δ” with capital “A”.

Authors' response: This typo has been fixed.

8. Same paragraph, what is meant by 28% of cells display “mostly” a single focus?

Authors' response: We are referring to the observation that in absence of exogenous DNA damage, most Rif1_{NTD}-GFP focus-positive cells contained only one Rif1 focus. We apologize for the unclear statement and have rephrased the text (page 12):

“(…), the majority of which contained a single Rif1_{NTD}-GFP focus (…).”

9. Discussion: “DNA damage-induced Rif1 focus formation was dependent on Pfa4…” (missing “Rif1”).

Authors' response: Text has been changed accordingly in the revised manuscript.

Reviewer #2 (Remarks to the Author):

Protein S-acylation plays critical roles in regulating protein localization, activity, stability, and complex formation. Hence, it is involved in various biological processes such as signal transduction, apoptosis, and metabolism. In this study, Fontana et al. made an interesting and important discovery that the S-acylation of Rif1 plays a key role in regulating DNA double-strand break repair. Overall, the manuscript was beautifully written, the experiments were well designed, and the data are solid. However, several minor issues need to be addressed.

Authors' response: We would like to thank reviewer 2 for the enthusiastic support. We have addressed the minor points raised as detailed below.

Page 6, Line 5. The conclusion that “protein S-palmitoylation is important for NHEJ efficiency in yeast” is an overstatement. Although Pfa4 is a palmitoyl acyltransferase, it might function in an S-palmitoylation-independent fashion. Moreover, protein S-palmitoylation was not investigated in the first section (i.e., The palmitoyl acyltransferase Pfa4 promotes NHEJ). Better to change the statement to “protein S-palmitoylation is potentially important for NHEJ efficiency in yeast”.

Authors' response: We agree, this statement is prematurely placed after section 1 of Results (see also reviewer 1, minor point 2). In the revised MS, the text on page 6 has been changed to read:

“These findings suggest that protein S-palmitoylation is important for NHEJ efficiency in yeast, and implicate Pfa4 and Rif1 in a common pathway of DSB repair pathway choice.”

Page 6, Results section #2. The authors did a nice job in prioritizing a list of Cys residues as candidate S- palmitoylation sites. But was the evolutionary conservation of these Cys residues across different species taken into consideration? If the authors believe that the S-palmitoylation of Rif1 is important in DNA double-strand break repair, the S-palmitoylation sites should be largely conserved from yeast to human.

Authors’ response: Rif1 primary amino acid sequences are highly divergent across organisms. Structural information for mammalian Rif1, which might allow to pinpoint surface-exposed cysteine residues within the HOOK domain that could play an equivalent role to yeast C466 and C473 identified in our study in budding yeast Rif1, is not available at present. There are several cysteines in mouse (<https://swisspalm.org/proteins/Q6PR54>) and human Rif1 (<https://swisspalm.org/proteins/Q5UIP0>) predicted to be palmitoylated, but it remains to be seen whether these sites or others are modified in cells, and if so, whether this impacts Rif1-mediated NHEJ in mammalian systems. We have added this information in the revised MS in Discussion, page 16:

“Of note, mammalian Rif1 S-palmitoylation is predicted by Swisspalm/CSS-Palm^{52,53}, but it remains to be determined whether these modifications occur in vivo and how they might relate to NHEJ.”

Pages 7 and 9. In page 7, the authors stated that “These results indicate an impairment of NHEJ by combined, but not individual, loss of potential Rif1 S-acylation sites C466 and C473.” However, in page 9, they concluded that “S- acylation occurs either at C466 or C473, but not – or very rarely – at both residues.” How can the findings be reconciled? The authors need to discuss this a bit.

Authors’ response: We agree that the notion that C466 or C473 may act as redundant acceptors of S-acylation to support Rif1’s role in NHEJ has not been made sufficiently clear. We have rephrased these statements in the revised MS. An extended para 1 on page 7 is now more explicit with regard to the results of single-site versus twin mutations at C466 and C473:

*“In contrast, cluster 2 mutants Rif1 C71A/C466A/C473A and Rif1 C466A/C473A were associated with reduced survival after DSB induction, phenocopying RIF1 (**Fig. 2b**) and/or PFA4 (**Fig. 1b**) deletions. Single-site mutants Rif1 C466A or C473A had no effect on cell survival after DSB formation. Consistent results were obtained upon chronic exposure of cells to Zeocin, where the rif1 C466A/C473A allele led to increased Zeocin resistance, similar to what is observed for rif1 Δ , pfa4 Δ , or rif1 Δ pfa4 Δ cells, while the rif1 C466A and rif1 C473A single-mutation alleles had no effect (**Fig. 2c**, see also **Fig. 1c**). All cluster 2 mutations had little or no impact on protein stability (**Supplementary Fig. 2c**). These results indicate an impairment of NHEJ by combined, but not individual loss of potential Rif1 S-acylation sites C466 and C473.”*

In addition, the text on page 10 was revised to read:

*“Peptides containing NEM-labeled C466 and C473 were detected, accounting for unmodified Rif1_{NTD}. Tryptic fragments containing either CAM-labeled C466 or CAM-labeled C473 were also detected, providing site-specific evidence for Rif1 S-acylation in vivo (**Fig. 4d**). Importantly, Rif1 peptides prepared from pfa4 Δ cells failed to incorporate CAM, confirming that the S-acylation of Rif1 at residues C466 or C473 is Pfa4-dependent. Alternative S-acylation at C466 and C473 might explain why only combined Rif1 C466A C473A mutations result in defective NHEJ, suggesting site redundancy for NHEJ-relevant S-acylation by Pfa4 in vivo.”*

Page 9, Line 17. Please replace “mass spectrometry” with “targeted mass spectrometry” or “parallel reaction monitoring”. In addition, the Methods sections for PRM development and application are not very clear. For example, how were the target peptides selected, and how were the PRM assays developed? In the “PRM data analysis” section, what was the purpose of performing Mascot database searching, and why was the palmitoyl (PALM; +238.2297) modification included in the list of the variable modifications? It seems that the authors either conducted a discovery proteomic analysis or attempted to quantify intact S-palmitoylated peptides, but probably had no success.

Authors’ response: We thank reviewer 2 for the suggestions, “mass spectrometry” has been replaced with “parallel reaction monitoring” on page 9. Our inclusion of the palmitoyl modification (PALM; +238.2297) is “historic”. We have tried detecting S-acylation directly in both biological material and on synthetic peptides. While we succeeded in detecting synthetic peptide palmitoylation, albeit with low sensitivity and high variability, we could not detect palmitoylation directly in biological material, consistent with published literature. We developed ACE to overcome precisely these limitations, greatly increasing sensitivity of detection and reproducibility. To avoid any confusion, we repeated the searches, excluding palmitoylation (PALM; +238.2297). Moreover, we have uploaded our results to the PRIDE proteomics data repository, where they can be accessed through username: reviewer56682@ebi.ac.uk; password: dbnUV59i. Please see also **Online methods (Supplementary data file**, section “Parallel Reaction Monitoring (PRM) data acquisition” from page 8) and **revised Supplementary Fig. 4** with new data panels.

Page 26, Figure 3c. Why were the BSA bands present?

Authors’ response: BSA was added to enhance protein stability and serves us to double-check equal protein loading on gels/in EMSA reactions. This information has been added in revised **Fig. 3** legend and the BSA band has been annotated on the protein gel in **Fig. 3c** (p. 28).

Page 32, Figure 6. What do the singly and doubly S-palmitoylated forms (above the left “Peripheral DSB”) stand for? In addition, the finding that “S-acylation occurs either at C466 or C473, but not – or very rarely – at both residues” needs to be taken into consideration in the illustration.

Authors’ response: We only depict mono-S-acylated Rif1 in the model figure (**now Fig. 7**) and this is now clearly stated in the revised figure legend to avoid any potential confusion where single S-acylation on closely spaced Rif1 molecules might give the impression of double-S-acylation. The legend to **Fig. 7** has been changed to read:

“In contrast, Pfa4-dependent Rif1 S-acylation at residues C466 or C473 (depicted as a zig-zag line) is essential for Rif1-mediated NHEJ.”

Reviewer #3 (Remarks to the Author):

In this paper, Fontana et al. build on three previously reported observations: that Rif1 is involved in DNA double strand break repair, that Rif1) is acylated and that this modification is mediated by the palmitoyltransferase Pfa4. The paper aims at connecting these events. While the paper is very well written, the novelty, the mechanistic insight is somewhat limited, and therefore the study appears preliminary.

The authors show that Rif1 can be S-acylated on two specific cysteines in the NTD but probably others as well that are not identified. The palmitoylation evidence should be strengthened. The main

conclusion that palmitoylated Rif1 mediates double strand break repair by accumulating at the inner nuclear membrane is insufficiently supported by the shown data. Given that the basis of the current findings had already been reported, more depth in mechanistic were expected.

Authors' response: We thank reviewer 3 for identifying the novelty of connecting S-acylation and DNA repair for the first time. We map *in vivo* S-acylation sites on Rif1 for the first time. Specifically, by mapping all those S-acylation sites that are essential for Rif1's role in promoting NHEJ, and by showing that S-acylation at these sites enables a nuclear-peripheral DNA damage response, we reveal a new fatty acylation-based mechanism for compartmentalized DNA repair. We appreciate the insightful comments and have addressed the points raised with substantial additional experimentation in the revised MS as detailed point-by-point below.

- The model proposed by the authors (Fig 6) indicates that S-acylated Rif1 localizes at the inner nuclear membrane and interacts with dsDNA. Where does the acylation by Pfa4 occur? In the cytoplasm? And, thus, would palmitoylated Rif1 enter the nucleus? Or does Pfa4 localize to the inner nuclear membrane? So far, the presence of palmitoyltransferases in the nucleus has not been reported, and thus this would need to be documented by localization studies of Pfa4 by fluorescence or electron microscopy, by subcellular fractionation (membrane vs soluble) to see Rif1 distribution in the different backgrounds.

Authors' response: We agree that the localization of Pfa4 is a relevant mechanistic question in the context of Rif1 S-acylation in DNA repair. Following this reviewer's suggestions, we have addressed the issue by fluorescence microscopy and subcellular fractionation (membrane vs soluble). The relatively recently discovered DHHC family of palmitoyl transferases are membrane proteins associated mostly with the endoplasmic reticulum (ER), the Golgi, sometimes the plasma membrane, and Pfa3 with the yeast vacuole. Pfa4, whose low expression level precludes direct detection, has been overexpressed in GFP-tagged form and shown to be ER-associated (Ohno et al. (2006). *Biochim Biophys Acta* 1761:474-483). Given the ER is continuous with the nuclear envelope, this places Pfa4 in a position that might allow access to cytoplasmic Rif1 shuttling into the nucleus (as suggested by Fox & Gartenberg (2012) *Nucleus* 3:251-255), or to nuclear Rif1 if Pfa4 can access lateral channels and traverse nuclear pores to reach the inner nuclear membrane; however, as pointed out by reviewer 3, currently no DHHC palmitoyl transferase has been shown to associate with the inner nuclear membrane and it is unknown whether Pfa4 has the ability to populate the different subdomains of the continuous ER-nuclear envelope membrane system. To tackle this question, we have now used a quantifiable cell-based assay, exploiting so-called theta nuclei (Deng & Hochstrasser (2006) *Nature* 443:827-831) to determine the ability of Pfa4 to localize to the inner nuclear membrane. The assay makes use of induced proliferation of the inner nuclear membrane following overexpression of nucleoporin Nup53. Inner nuclear membrane proliferation leads to the formation of membrane structures packed against the nuclear envelope, which, in 20-30% of cells, have been shown to cut across the nuclear interior (Marelli et al. (2001) *J. Cell Biol.* 153:709-723; Deng & Hochstrasser (2006) *Nature* 443:827-831). These transecting inner nuclear membrane structures lead to theta nuclei with an appearance reminiscent of the Greek letter theta. Theta nuclei can be quantified on the basis of GFP-tagged candidate proteins with the highly selective ability to localize to the inner nuclear membrane. As expected (Deng & Hochstrasser (2006) *Nature* 443:827-831), we found upon Nup53 overexpression that ER membrane protein Sec61-GFP efficiently accessed the inner nuclear membrane, lighting up Nup53-induced theta nuclei. In contrast, cells expressing ER membrane protein Hrd1-GFP showed 4-fold lower levels of fluorescent theta nuclei, consistent with poorer inner nuclear membrane access (Deng & Hochstrasser (2006) *Nature* 443:827-831). Pfa4-GFP showed an intermediate phenotype, localizing to more than half the theta nuclei compared to the Sec61-GFP positive control. These data show, for the first time, that palmitoyl transferase Pfa4 associates with the inner nuclear membrane, providing access to nuclear

S-acylation targets. We next treated cells with Zeocin to induce DNA damage and monitored inner nuclear membrane association. The localization of Sec61-GFP or Hrd1-GFP to theta structures did not change, indicating that Zeocin did not impact access of Sec61 or Hrd1 to the inner nuclear membrane. Importantly, Pfa4-GFP had increased access to the inner nuclear membrane after DNA damage, now marking theta structures above the level of the Sec61-GFP positive control. These data show that nuclear Rif1 is accessible to Pfa4 and suggest that nuclear Rif1 S-acylation may be facilitated by increased Pfa4 at the inner nuclear membrane upon DNA damage. This provides a conceptually satisfying pathway for Pfa4-dependent S-acylation of nuclear Rif1. This important new mechanistic insight is presented in the revised MS in **new main Figure 5**. In addition, we performed subcellular fractionation (membrane vs soluble) experiments to determine Rif1 distribution from cells expressing wild-type Rif1, Rif1 palmitoylation mutation C466A C473A, and wild-type Rif1 in absence of Pfa4. Previously, Fox and co-workers (Park et al. (2011) Proc Natl Acad Sci 108:14572-14577) used fractionation experiments to show that Rif1 was enriched in the insoluble pellet fraction of extracts from wild-type relative to *pfa4Δ* extracts. At the same time, Rif1 levels were decreased in the soluble fraction of *PFA4* wild-type relative to *pfa4Δ* extracts. We have now recapitulated and extended these results. As shown in **new Supplementary Figure 5**, the palmitoylation mutant of Rif1 (C466A C473A) exhibited an intermediate phenotype, being mildly enriched in the soluble fraction and diminished in the insoluble pellet fraction compared to wild-type Rif1 under unperturbed and DNA damage conditions. Overall, the fractionation and Pfa4-localization results are consistent with Pfa4-dependent S-acylation anchoring Rif1 to the inner nuclear membrane and with S-acylation at C466 and C473 – the residues essential for Rif1-mediated NHEJ and DNA end-protection – contributing directly to Rif1 anchoring. A new section has been included in the MS, starting on page 10:

“To address the question how S-acylation may promote Rif1-mediated NHEJ, we first sought to determine whether the modifying enzyme, Pfa4, has access to the cell nucleus. Although membrane associated, palmitoyl transferases have so far not been observed at the inner nuclear membrane. Pfa4-GFP has been localized at the endoplasmic reticulum (ER)⁵⁶, which is continuous with the nuclear envelope, but inner nuclear membrane access is selective, and whether Pfa4 can populate this sub-compartment is unknown (Fig. 5a). Taking advantage of induced inner nuclear membrane proliferation following overexpression of nucleoporin Nup53, we asked whether Pfa4-GFP can access the resulting, distinctive membrane structures⁵⁷. These intranuclear lamellae have been shown to present in the form of so-called theta (θ) nuclei with transecting membranes, providing the basis for a quantifiable fluorescence assay for testing inner nuclear membrane localization of GFP-tagged candidate proteins⁵⁸. As a positive control, we expressed ER membrane protein Sec61-GFP, which accessed the inner nuclear membrane, efficiently decorating θ structures⁵⁸ induced by Nup53 overexpression (Fig. 5b). As expected, cells expressing ER membrane protein Hrd1-GFP showed ~4-fold lower levels of fluorescent θ nuclei compared to Sec61-GFP (Fig. 5b), reflecting poorer inner nuclear membrane access⁵⁸. Pfa4-GFP showed an intermediate phenotype, populating theta nuclei at ~1.5-fold lower levels than Sec61-GFP (Fig. 5b). Upon DNA-damage treatment with Zeocin, the localization of Sec61-GFP or Hrd1-GFP to θ structures did not change, while Pfa4-GFP-associated θ structures increased ~2-fold. These data show that Pfa4 localizes to the inner nuclear membrane in unperturbed conditions, and this localization is enhanced after DNA damage (Fig. 5b). Using cell fractionation, we confirmed previous results⁴² indicating Pfa4-dependent membrane associations of Rif1, which proved partially dependent on Rif1 residues C466 and C473 (Supplementary Fig. 5). Together, these results are consistent with Pfa4 having access to Rif1 in the nucleus, where NHEJ-relevant S-acylation at C466 and C473 may contribute to Rif1-membrane interactions.”

- ABE results show decreased but not completely abolished signal upon mutation of cysteines or Pfa4 deletion. Identification of all palmitoylation sites would be important. Also data that excludes the other palmitoyltransferases would strengthen the conclusion, excluding indirect effects of Pfa4.

The ABE or derivatives thereof are not ideal to study proteins with multiple palmitoylation sites since the presence of a single remaining site is in principle sufficient to bring down the protein. So, alternative methods should be used to confirm these findings.

Authors' response: We thank reviewer 3 for the suggestion to test the potential involvement of palmitoyl-transferases other than Pfa4 in the NHEJ-relevant S-acylation of Rif1. Defects in Rif1-mediated NHEJ result in the tell-tale phenotype of increased Zeocin resistance, as observed for *rif1Δ*, *pfa4Δ*, and *rif1* C466A C473A mutants (**Fig. 1c** and **Fig. 2c**). We have now performed Zeocin-resistance assays in strains individually deleted for each of the seven DHHC palmitoyl-transferases present in budding yeast (Akr1, Akr2, Erf2, Pfa3, Pfa4, Pfa5, Swf1). Importantly, only deletion of *PFA4* phenocopied *rif1Δ*, leading to increased Zeocin resistance indicative of NHEJ defects. This is in line with Pfa4 being associated with the inner nuclear membrane (**new Fig. 5**) and supports Rif1 as a direct Pfa4 S-acylation target. These important data are now included in the revised MS as **new Supplementary Fig. 1b**, referred to on page 5 of the revised MS:

"We have shown previously that cells deleted for RIF1 exhibit a ~2-fold increase in Zeocin resistance compared to wild-type control cells⁸. A similar increase in survival upon Zeocin exposure was observed for cells deleted for PFA4, but not cells deleted for any of the other six palmitoyl-transferases present in budding yeast (Supplementary Fig. 1b). Furthermore, Zeocin resistance levels of rif1Δ pfa4Δ double-mutant cells was no greater than that of rif1Δ or pfa4Δ single mutant cells (Fig. 1c). These results suggest that Rif1 and Pfa4 may act jointly to facilitate NHEJ."

ABE assays do have the above-mentioned limitations, and we therefore developed ACE. In ACE, enrichment of the target protein is not based on protein modifications, and, being a mass spectrometry-based method, ACE provides direct information whether a particular amino acid is modified or not, allowing us to conclude that Rif1 C466 or C473 are S-acylated *in vivo* in a Pfa4-dependent manner. While the ABE results and our new subcellular fractionation data (**Fig. 4b**; see also **new Supplementary Fig. 5**) suggest that Pfa4 may be able to mediate S-acylation on Rif1 C466A C473A mutants to some extent, importantly, these modifications cannot support Rif1-mediated NHEJ in absence of C466/C473 S-acylation. Having also demonstrated that S-acylation at C466 or C473 is essential for Rif1-mediated NHEJ, and that mutating all other Rif1 cysteines with a potential role in promoting NHEJ was inconsequential (**Fig. 2**), we feel that determining potential additional S-acylation sites within Rif1 is not critical for demonstrating Rif1 S-acylation as a novel post-translational modification regulating NHEJ and the conclusions drawn in the current study.

- Mass spectrometry was used to analyze palmitoylation. While this technique is elegant, the conclusion that the two sites cannot be simultaneously modified would need confirmation by other methods such as PEGylation (a method established by various groups including that of Fukata and that of Huang), which allows an estimation of the site occupancy and also what percentage of Rif1 is actually modified in the cell.

Authors' response: In the revised MS, we have extended our analysis of the four different forms of the synthetic Rif1 peptide spanning residues 463-479 (C466/C473 unmodified, single modification at C466 or C473, double-modified). The MS2 spectra (in particular fragments y3 and y9-y12) and retention times in HPLC of these peptides are very similar (see **new Supplementary Fig. 4b, c**). Despite this, signals for double-modifications at Rif1 C466 and C473 are virtually indistinguishable from noise in biological samples (see **Source data file** and **data repository PXD012137** uploaded to PRIDE, accessible with username: reviewer56682@ebi.ac.uk; password: dbnUV59i) and are omitted from revised **Fig. 4** for clarity. This is based on a more extensive analysis of the biological material in the revised MS, allowing improved inter-peptide comparisons. We exclusively integrated the signals from y ions y3 and y9-y12 for the four different peptide species detected (see **new Supplementary**

Fig. 4b, c). Results for biological samples were further normalized with five non-modified Rif1 peptides (shown in **Supplementary Table 1** and PRM analysis shown in the **Source data file**). We conclude from this data with reasonable confidence that twin-modified Rif1 with S-acylation on C466 and C473 is likely to be absent or very rare, notwithstanding that a herein undetected species of Rif1 with twin S-acylation at C466 and C473 might exist. It is important to note that even if evidence of a twin modification of Rif1 at C466 and C473 could be produced, this information would be of little or no relevance for the capacity of Rif1 to mediate NHEJ as shown by our functional repair assays demonstrating that single alanine substitutions at C466 or C473 do not result in a NHEJ defect, whereas mutating both sites together results in a NHEJ defect. Hence, we conclude that “(...) at least one Pfa4-dependent S-acylation event at C466 or C473 is essential for Rif1-mediated NHEJ” (p. 14). Based on our improved analysis of the biological samples, we estimate that about 10% of Rif1 is S-acylated at either C466 or C473, indicating a total fraction of 15-20% of Rif1 being S-acylated at the NHEJ-relevant sites C466/C473. This information has been added on page 10 of the revised MS:

“The PRM values for double NEM-modified Rif1 peptides spanning residues 463 to 479 are approximately 10 times higher than the values for peptides derived from Rif1 S-acylated at C466 or C473. This implies that a substantial fraction of 15-20% of Rif1 was S-acylated at either C466 or C473 in wild-type cells (Fig. 4d, see also Supplementary Fig. 4).”

- As mentioned by the authors in the manuscript, palmitoylation is reversible and can be dynamic. The dynamic aspect is missing in the current work. The key question is raised by the authors in their conclusion; what is the “dynamic regulation of Rif1 by Pfa4 in response to DNA damage”. Is the acylation status of Rif1 modified upon induction of DNA damage? This should be address in detail, looking at the different palmitoylation sites. The mass spec and the ACE analyses were done it seems on untreated cells.

In the revised MS, we have carried out new ACE experiments to assess Rif1 C466/C473 S-acylation in untreated cells and cells treated with radiomimetic drug Zeocin in parallel (**Fig. 4, new panel e**). Our results (**data repository PXD012137** at PRIDE, username: reviewer56682@ebi.ac.uk; password: dbnUV59i) indicate that Rif1 is constitutively S-acylated and suggest that DNA damage does not lead to a dramatic change of global Rif1 S-acylation. This does of course not rule out local regulation of Rif1 S-acylation dynamics in response to DNA-damage (see also ref. 1, minor point 5), which will be the subject of future investigations that are beyond the scope of the current study. Textual changes to the revised MS on page 10 are as follows:

“DNA-damage treatment with Zeocin did not lead to gross changes in Rif1 S-acylation levels (Fig. 4e), consistent with constitutive Rif1 S-acylation.”

- Statistical analysis is missing in figures 1 to 4. T-test-associated p-values should be provided for all discussed comparisons. Also, statistical analysis shown in Figure 5 seems inconsistent with the conclusions drawn from the data. The authors should provide the p-values associated to the comparisons discussed in the text. Additionally, the 300 cells obtained from 3 experiments cannot be processed in the same way for statistical purposes as they are not truly independent.

During this revision, we have increased the numbers of observations for the majority of panels in **Figures 1-4 (Fig. 1, panels b-d, Fig. 2c, Fig. 3, panels a and b)** and performed appropriate statistical tests (p-values indicated) for each experimental setup (please refer to the respective **Figure legends** and to **Online methods** for a description of the statistical tests used). We also performed a new analysis of the data previously depicted in **Fig. 5 (now revised Fig. 6, panels b and c)**. Following this reviewer’s and reviewer 1’s suggestions, we present a direct comparison of the fraction of Rif1

focus-positive cells in wild-type cells, and cells expressing the Rif1 C466A C473A S-acylation mutant, the Rif1 HOOK DNA-binding mutant, and also a new Rif1 K437E K563E K570E C466A C473A (S-acylation/HOOK) mutant before and after DNA damage treatment, with added statistical analysis for inter-strain comparisons. For **revised Fig. 6b**, we considered the three independent experiments underlying the 300 cells analyzed (each experiment approx. 100 cells; data provided in the accompanying **Source data file**), calculating % of focus-positive and focus-negative cells for each set. The mean, standard deviation and S.E.M. for the three experiments were calculated, and data were plotted with statistical analysis as detailed in the **Figure legend** and **Online methods**. In **Fig. 6c**, the 300 cells are presented as one pool for clarity while the three independent experiments of approx. 100 cells underlying the graph were used to calculate the average amount of Rif1 foci per cell, which also formed the basis to assess experimental variability and calculate statistical significance (p-values indicated).

Textual changes regarding **Fig. 6** are as follows (starting on page 11):

*“In untreated conditions, we observed nuclear Rif1_{NTD}-GFP foci in ~28% of cells. After DNA-damage treatment with Zeocin or ionizing radiation (IR), focus formation was strongly induced, reaching a peak ~30 min post-treatment (**Supplementary Fig. 6b**), when ~60% (Zeocin) and ~80% (IR) of cells exhibited Rif1_{NTD}-GFP foci (**Fig. 6a, b** and **Supplementary Fig. 6c, d**). Moreover, while the majority of focus-positive cells in unperturbed conditions contained a single Rif1_{NTD}-GFP focus, most focus-positive cells exhibited multiple (up to four) foci after DNA damage treatment (**Fig. 6c**). DNA damage-induced Rif1_{NTD}-GFP foci were observed in G1 and S/G2 cells with no overt cell-cycle dependence. DNA-damage treatment did not lead to increased Rif1_{NTD} expression levels (**Supplementary Fig. 6e**), suggesting that focus formation reflected the redistribution of Rif1_{NTD}-GFP into foci upon DNA damage.”*

*Next, we analyzed Rif1_{NTD}-GFP foci in pfa4Δ cells. Untreated cells were indistinguishable from wild-type with ~30% Rif1_{NTD}-GFP focus-positive cells, the majority of which contained a single Rif1_{NTD}-GFP focus (**Fig. 6b, c** and **Supplementary Fig. 6c, d**). In contrast, the ability to form DNA damage-induced Rif1_{NTD}-GFP foci (~36% and ~38% focus-positive cells after Zeocin and IR, respectively, see **Fig. 6b** and **Supplementary Fig. 6d, f**) and the formation of multiple Rif1 foci in response to DNA damage (**Fig. 6c**) was significantly abrogated in pfa4Δ cells. Protein levels of Rif1_{NTD} remained unchanged upon loss of Pfa4 (**Supplementary Fig. 6g**). Importantly, in Pfa4-proficient cells, introducing the S-acylation mutation C466A/C473A also diminished the formation of DNA damage-induced Rif1_{NTD} foci and the ability of cells to form multiple Rif1 foci (**Fig. 6b, c**). Like S-acylation mutant Rif1_{NTD} C466A/C473A, the Rif1 HOOK DNA-binding mutant was strongly compromised in its ability to form foci in response to DNA-damage treatment (**Fig. 6b, c**). Combining the S-acylation and HOOK mutations led to a more severe phenotype compared to either the Rif1_{NTD} C466A/C473A or the Rif1 HOOK mutant (**Fig. 6b, c**). Thus, Pfa4-dependent S-acylation of Rif1 at C466/C473 and the ability of Rif1 to bind DNA contribute to effective Rif1 accumulation upon DNA damage.”*

- Fig 5c: the distribution of Rif1 signal between the three defined zones should be compared to the non-treated condition. How would this distribution look in the Pfa4-deletion or the Rif1 mutant backgrounds?

Authors' response: In the revised MS, we have greatly extended our zoning analyses (see also reviewer 1, point 4). The analysis now includes Rif1 in a pfa4Δ background, the Rif1 C466A C473A S-acylation mutant, the Rif1 HOOK DNA-binding mutant, and a newly created, combined Rif1 HOOK/S-acylation mutant (Rif1 C466A C473A K437E K563E K570E). All these mutant cells are greatly impaired in the formation of DNA damage-induced Rif1 foci (**revised Fig. 6a, b** and **Supplementary Fig. 6c, d**). Since we focus on the newly discovered DNA-damage induced Rif1 foci, which are distinct from

previously described telomeric Rif1 signals, we had initially concentrated our description of the subnuclear localization of DNA-damage induced foci on wild-type cells, finding a strong bias for peripheral localization (zone 1) in zoning assays. Residual Rif1 foci in all mutants analyzed are mostly those that are observed spontaneously in all strains and these are, by definition, not DNA-damage induced foci. Nevertheless, any residual Rif1 focus-formation induced upon DNA damage in the S-acylation-compromised mutants (including *pfa4Δ* and cells expressing Rif1 C466A C473A and Rif1 C466A C473A K437E K563E K570E) exhibited a greater proportion of foci situated more distant to the inner nuclear membrane in zone 2. One exception was the Rif1 DNA-binding HOOK mutant with intact S-acylation sites C466/C473. These new data (**Figure 6, new panel d**) showing that zone 1-localization is abrogated whenever S-acylation of Rif1 is compromised are consistent with Pfa4-dependent S-acylation at C466 and C473 promoting Rif1 sequestration at the inner nuclear membrane. While we find a very similar behavior for all strains tested for spontaneous Rif1 foci that appear independently of DNA damage treatment (**see Figure A below**), in the revised MS we present zoning assays after DNA damage, given the focus on the DNA damage-induced component of Rif1 focus formation. The text starting on page 12 has been revised to read:

*“To determine the sub-nuclear localization of Rif1_{NTD}-GFP foci, we scored their position relative to the nuclear envelope marked by fluorescently tagged nuclear-pore component Nup49⁵⁹. Dividing the nucleus into three concentric zones of equal area, we found a strong bias of Rif1 accumulation in outermost zone 1, at the nuclear periphery, in wild-type cells. Upon DNA damage, ~60% of Rif1_{NTD}-GFP foci localized in zone 1 (**Fig. 6d**). While cells expressing the Rif1 HOOK DNA-binding mutant maintained a strong localization bias to zone 1, compromised Rif1 S-acylation in cells expressing Rif1_{NTD} C466A/C473A, Rif1_{NTD} C466A/C473A HOOK, or wild-type Rif1_{NTD} in a *pfa4Δ* background, led to an increase in zone 2-localized foci at the expense of zone 1-localized foci. Thus, Rif1 S-acylation mutants display an apparent reduction in Rif1-inner nuclear membrane interactions in conjunction with a significant impairment in the formation of DNA damage-induced Rif1-foci observed in all mutant backgrounds tested (**Fig. 6b, c and Supplementary Fig. 6c, d, and f**). Taken together, these data are consistent with a model where enrichment of Rif1 at the inner nuclear membrane mediated by Pfa4-dependent S-acylation of C466/C473 and its intrinsic DNA-binding activity enable effective Rif1 accumulation at nuclear-peripheral DNA damage, promoting preferential repair of membrane-proximal DSBs along the NHEJ pathway (**Fig. 7**).”*

Figure A. A zoning assay scoring the position of Rif1_{NTD}-GFP foci in the indicated strains in unperturbed conditions. Analysis performed as described for **Main text Figure 6d** in the revised MS.

Minor comments:

- It is not clear why the authors resort to DTT to cleave the thioester bonds in their ACE method instead of using hydroxylamine as for ABE. Also, the MS data provided in Fig S4 supporting TCEP does not cleave thioester bonds is not convincing.

Authors' response: We have used hydroxylamine for ABE experiments, following well established protocols (Wan et al. (2007). *Nature Protocols*, 2:1573–1584). In ACE experiments, we switched to DTT for thioester bond cleavage to optimize buffer compatibility with mass spec analyses. DTT had already been described as a reagent that efficiently removes cysteine S-acylation (Ji et al. (2013). *Analytical Chemistry* 85: 11952-9). To verify suitability for our purposes further, we have now expanded **Supplementary Fig. 4** to include two additional synthetic, palmitoylated peptides on which we performed reduction with DTT vs TCEP. As illustrated in the revised **Supplementary Fig. 4, panel a**, alkylation with IAC or NEM occurs only when peptides modified with S-palmitoylation were treated with DTT, but not with TCEP. This shows that TCEP does not cleave these thioester bonds, in line with an existing report using synthetic peptides (Ji et al. (2013). *Analytical Chemistry* 85: 11952-9). We confirmed the findings by Ji and co-workers and expanded the technique to protein extracts in our ACE method (see **Fig. 4** and **Online methods**). The robustness of our ACE protocol is perhaps best demonstrated by the fact that no CAM-labeling of Rif1 is detected at C466 and C473 in strains deleted for S-acyltransferase Pfa4 while NEM-modified C466 and C473 are readily detected in both wild-type and *pfa4*Δ (please see **Fig. 4**).

- It would be nice to see an example of the raw data used for the plots provided in Figs 1, 3, and S4.

Authors' response: This data is provided in a new **Source Data File** that accompanies the MS.

- Catalog numbers (or even better RRID) should be provided for all antibodies used, and ideally other reagents (as those used in ABE/ACE methods).

Authors' response: Catalog numbers and RRID for all antibodies and catalog numbers/manufacturer information for all reagents are now provided in the **Online Methods** in the revised version of the MS.

- Using the label “RIF1” to refer to the WT strain in Fig 1 is confusing. Similarly, labels “Rif1NTD in delta-Rif1” in Fig 3a is misleading.

Authors' response: We agree, labeling had been not ideal, this was changed throughout the figures according to this reviewer's suggestions.

Reviewers' comments:

Reviewer #1 (Remarks to the Author):

The authors have adequately addressed the concerns of this reviewer.

Reviewer #2 (Remarks to the Author):

Fontana et al. made significant effort to address the concerns and improved the manuscript. However, several issues still need to be addressed before the manuscript can be accepted.

1. Although the effort of applying PRM to quantify S-acylated peptides is applaudable, the PRM assay was not rigorously designed. For example, what are the limit of detection (LOD), the limit of quantification (LOQ), and the linear range? Without such information, it is hard to tell whether there is really significant difference between WT and *pfa4Δ* for CAM-modified peptides in Fig. 4d. In addition, please note that, to obtain these pieces of information, the matrix effect needs to be taken into consideration. This is usually handled by spiking heavy peptide standards into samples.
2. Please use S-acylation consistently unless intact S-palmitoylated peptides were measured in biological samples (which was not successful in the study).
3. The S in the "S-acylation" needs to be italicized.
4. Figure 7. Because the Pfa4 was the only S-acylation enzyme for Rif1 and was found to be at least partly localized in the inner nuclear membrane, don't the authors think that Pfa4 should be included in the working model?
5. Supplemental Figure 4a. The MS1 intensities (y axis) were original and not log transformed.

We thank the reviewers for their support during the revision of our manuscript “*Rif1 S-acylation mediates DNA double-strand break repair at the inner nuclear membrane*”. We were very pleased to learn of their endorsement for publication and have addressed remaining points as detailed below. (Textual changes in the revised MS are highlighted green.)

Reviewers' comments:

Reviewer #1 (Remarks to the Author):

The authors have adequately addressed the concerns of this reviewer.

Authors' response: We thank reviewer 1 for a constructive revision process.

Reviewer #2 (Remarks to the Author):

Fontana et al. made significant effort to address the concerns and improved the manuscript. However, several issues still need to be addressed before the manuscript can be accepted.

Authors' response: We thank reviewer 2 for previous comments on our work on which our major experimental revision was based and appreciate the acknowledgment of the improvements to the MS that have arisen as a consequence. The issues raised herein have been addressed as detailed in the point-by-point response below.

1. Although the effort of applying PRM to quantify S-acylated peptides is applaudable, the PRM assay was not rigorously designed. For example, what are the limit of detection (LOD), the limit of quantification (LOQ), and the linear range? Without such information, it is hard to tell whether there is really significant difference between WT and *pfa4Δ* for CAM-modified peptides in Fig. 4d. In addition, please note that, to obtain these pieces of information, the matrix effect needs to be taken into consideration. This is usually handled by spiking heavy peptide standards into samples.

Authors' response: We agree that an accurate quantification of absolute amounts of NEM and CAM-modified Rif1 peptides spanning amino acids 463-479 and their relative stoichiometry in wild-type and *pfa4Δ* cells would necessarily entail measurements with tailored, stable isotope-labeled synthetic peptides. In this study, we developed the ACE method with a different purpose in mind: to provide site-specific information on Rif1 S-acylation. ACE combined with label-free PRM measurements fulfills this purpose, allowing us to detect Rif1 S-acylation at C466 and C473. This validates a series of experiments in the paper providing evidence for an involvement of C466/C473 S-acylation in Rif1-mediated NHEJ: We first show that loss of Pfa4 (but not any other DHHC palmitoyl-transferases, **Supplementary Fig. 1**) results in NHEJ defects that phenocopy—and are epistatic with—loss of Rif1 (**Fig. 1**). Next, we map C466 and C473 by mutational analysis as the only potential S-acylation sites essential for Rif1-mediated NHEJ (**Fig. 2**), coinciding with a prediction of C466 being a palmitoylation target by Swisspalm (**Supplementary Fig. 2**). Importantly, point mutations C466A and C473A do not affect Rif1's ability to bind DNA, yet lead to de-protection of DNA double-strand break ends, further supporting the requirement of Pfa4-modified post-translational modification by S-acylation specifically at C466 and C473 for Rif1-mediated DNA end-protection and NHEJ (**Fig. 3, Supplementary Fig. 3**). In **Fig. 6** (and **Supplementary Fig. 5 and 6**), we show by zoning and biochemical assays that Pfa4 and Rif1 C466 and C473 are required for a nuclear-peripheral DNA damage response, consistent with Pfa4-dependent S-acylation of Rif1 at C466 and C473 providing membrane anchors to sequester Rif1 at the inner nuclear membrane, where it is poised to respond to DNA damage locally (**Fig. 7**). In **Fig. 4**, we used an ABE approach followed by Western blotting to provide evidence that S-acylated Rif1 is strongly reduced in absence of Pfa4, which is consistent with previously published findings of Pfa4-dependent Rif1 S-

acylation (Park et al. (2011) Proc Natl Acad Sci 108:14572-14577), and is also reduced for the Rif1 C466A/C473A mutant, consistent with the notion that S-acylation occurs at these sites. Against this background, we devised ACE in order to provide additional, site-specific evidence for Rif1 S-acylation at C466 and C473. In perfect agreement with our genetic, cytologic, and biochemical analyses, the ACE-PRM data clearly demonstrate S-acylation of C466 and C473 in vivo by identifying the corresponding chemically modified Rif1 peptides.

The peptides indicative of Rif1 C466/C473 S-acylation could be detected in wild-type cells—and while not determined experimentally for these specific peptides—we expect the LOD to be at least an order of magnitude and more likely 50-100 times lower than the measured PRM signals at 120000 resolution in the ORBITRAP Fusion device used for our analysis. Yet, peptides indicative of Rif1 C466/C473 S-acylation were not detected for Rif1 prepared from *pfa4Δ* cells. To ensure that intra- and inter-sample comparison would not be skewed due to fluctuations in target protein content, we used measurements of five unmodified tryptic Rif1 peptides to normalize for Rif1 amounts between samples. These measurements show that very similar amounts of Rif1 were consistently present across samples and conditions, providing confidence in our label-free measurements. To be clear on this point, we have now included the PRM values obtained from these control peptides, their sequence and position within Rif1 in **revised Fig. 4** (panels **d** and **e**). With regard to the NEM and CAM-modified peptides, it is important to note they behave similarly in LC (retention time) and MS (MS1 charge distribution and MS2 (HCD fragmentation)), please see **Supplementary Figure 4b,c**. Of these, Rif1 peptide 463-479 with NEM-labeled C466 and C473 (IYQC[Nem]IMLSPVC[Nem]ETIPEK) shows virtually the same distribution across all conditions and strain backgrounds (wild-type or *pfa4Δ*). In contrast, signals for the C466-NEM/C473-CAM (IYQC[Nem]IMLSPVC[Cam]ETIPEK) and the C466-CAM/473-NEM (IYQC[Cam]IMLSPVC[Nem]ETIPEK) peptides were detected (well above the noise) at similar levels in 9 independent wild-type samples (**Fig. 4d,e**), yet not in 3 independent experiments with *pfa4Δ* cells (**Fig. 4d**). Since we analyze substantially similar samples similar ion suppression effects can be expected. Our findings therefore clearly support the role of Pfa4 in the S-acylation of Rif1 at sites C466 and C473. Despite the black-and-white nature of our results, we consciously did not make a statement on a “significant difference between WT and *pfa4Δ* for CAM-modified peptides in Fig. 4d” in absence of label-based measurements of absolute amounts of the identified peptides. However, we feel that our label-free analyses, along with experimental evidence from multiple other approaches and published data (Park et al. (2011) Proc Natl Acad Sci 108:14572-14577), contribute greatly not only to demonstrating C466/C473 S-acylation of Rif1, but also in providing compelling and highly reproducible evidence that Pfa4 mediates the modification of Rif1 at these NHEJ-relevant sites. Both, ABE and ACE (**Fig. 4**) show that Rif1 S-acylation is perturbed in absence of Pfa4, and the body of evidence presented throughout our study is best explained by Rif1 C466/C473 being NHEJ-relevant, Pfa4-dependent S-acylation targets. To steer clear of any potential overstatement regarding the data presented in **revised Fig. 4d**, we have amended the corresponding passage in the text and the revised MS on p. 10 now reads:

*“Immuno-precipitated Rif1_{NTD} was subjected to parallel reaction monitoring (PRM), analyzing NEM and/or CAM-labeled C466 and C473 in tryptic peptide fragments spanning residues 463 to 479. Peptides containing NEM-labeled C466 and C473 were detected in wild-type and *pfa4Δ* cells, accounting for unmodified Rif1_{NTD}. Tryptic fragments containing either CAM-labeled C466 or CAM-labeled C473 were detected in wild-type cells, providing site-specific evidence for Rif1 S-acylation in vivo (Fig. 4d). Under these conditions, CAM-modified Rif1 peptides from *pfa4Δ* cells were not detected, supporting the role of Pfa4 in Rif1 C466/C473 S-acylation.”*

As mentioned above, we have revised Fig. **4d, e**, adding control peptides and now showing integrated PRM counts throughout. The data are shown in logarithmic scale (see also point 5 below), and this is stated in the revised Figure legend, p. 31:

*“(d) Mass-spectrometric analysis of tryptic Rif1 fragments spanning amino acids 463 to 479. Following ACE, Rif1_{NTD} tryptic peptides were subjected to parallel reaction monitoring (PRM), measuring NEM (unmodified Rif1) and/or CAM-labeled (reflecting in vivo S-acylation) C466 and C473 in tryptic fragments spanning residues 463 to 479 in wild-type vs. pfa4Δ. Integrated PRM counts are presented as mean values ± s.e.m. and were normalized using measurements of the five non-modified Rif1 peptides shown on the right (see **Supplementary Table 1** for additional information). Data are shown in logarithmic scale (n = 3 independent experiments). (e) Measurements of C466/C473 NEM and/or CAM-labeled peptides (left panel) and unmodified control peptides (right panel) of Rif1 in untreated vs. Zeocin-treated wild-type cells. PRM analysis of Rif1_{NTD} peptides as in panel d. Mean values of integrated PRM counts ± s.e.m (n = 3 independent experiments) are shown in logarithmic scale.”*

2. Please use S-acylation consistently unless intact S-palmitoylated peptides were measured in biological samples (which was not successful in the study).

Authors' response: We agree that S-acylation and S-palmitoylation cannot be used synonymously and have reviewed our use of these terms throughout the MS. In the main MS file (excl. the bibliography), there are 71 mentions of “S-acylation” compared to 9 mentions of S-palmitoylation (listed below). In 8 cases, we refer to the process of S-palmitoylation by Pfa4, prediction of S-palmitoylation sites by Swisspalm, possibilities of detecting S-palmitoylation, and generally to S-palmitoylation as a post translational modification. A ninth case was ambiguous, and we exchanged S-palmitoylation for S-acylation in this instance.

(#1, p. 4) *“Here, we show that Pfa4-dependent S-acylation of Rif1 promotes the attenuation of DNA end-resection and DSB repair by NHEJ, implicating protein S-palmitoylation in DNA repair.”*

(#2, p. 6) *“These findings suggest that protein S-palmitoylation is important for NHEJ efficiency in yeast, and implicate Pfa4 and Rif1 in a common pathway of DSB repair pathway choice.”*

(#3, p. 6) *“Notably, in silico analysis using Swisspalm/CSS-Palmu^{52,53} predicted one of the cysteines that we selected, C466, as a residue of potential Rif1 S-palmitoylation (**Supplementary Fig. 2b**).”*

(#4, p.6) *“Cysteine-acyl thioesters were then cleaved by treatment with dithiothreitol (DTT) and modified with CAM, allowing the identification of S-palmitoylation sites in peptides with high sensitivity (see **Supplementary Fig. 4b, c**).”*

(#5, p.14) *“We demonstrate that C466 or C473 are modified by S-acylation in vivo in a strictly Pfa4-dependent manner, strongly suggesting they are direct Pfa4 S-palmitoylation targets (**Fig. 4**).”*

(#6, p.16) *“Of note, mammalian Rif1 S-palmitoylation is predicted by Swisspalm/CSS-Palm^{52,53}, but it remains to be determined whether these modifications occur in vivo and how they might relate to NHEJ.”*

(#7, p.16) *“Interestingly, chemical inhibition of protein S-palmitoylation in mammalian cells led to a muted DNA-damage response⁷⁶.”*

(#8, p.16) *“The reversible nature of S-palmitoylation is reminiscent of well-established posttranslational modifications with important roles in DSB repair, including protein phosphorylation and ubiquitination⁷⁷, potentially allowing the dynamic regulation of Rif1 at DNA damage.”*

(#9, p.16) *“The essential requirement for S-palmitoylation in Rif1-mediated NHEJ provides the first example of a direct involvement of fatty acylation in DNA repair.”*

Changed to: *"The essential requirement for S-acylation in Rif1-mediated NHEJ provides the first example of a direct involvement of fatty acylation in DNA repair"*.

In the Supplementary Data file, we speak of S-palmitoylation only in the context of site prediction and synthetic palmitoylated peptides.

3. The S in the "S-acylation" needs to be italicized.

Authors' response: "*S-acylation*" has been italicized throughout the MS main text and Supplementary Data file (this change not highlighted in the revised MS).

4. Figure 7. Because the Pfa4 was the only S-acylation enzyme for Rif1 and was found to be at least partly localized in the inner nuclear membrane, don't the authors think that Pfa4 should be included in the working model?

Authors' response: We agree and have included an updated model depicting Pfa4 in **revised Figure 7**.

5. Supplemental Figure 4a. The MS1 intensities (y axis) were original and not log transformed.

Authors' response: We thank reviewer 2 for pointing this out. The MS1 intensities are original but shown in log scale. To make this point clearer we deleted "log" from the Y axis in **revised Supplementary Figure 4** and added this information to the Figure legend, p. 18 of the Supplementary Data File:

"The MS1 intensities are shown in log scale".

For consistency, a similar way of illustrating the data was adopted in **revised Figure 4d, e**.

REVIEWERS' COMMENTS:

Reviewer #2 (Remarks to the Author):

The authors have addressed all my concerns and comments. I recommend acceptance of the manuscript.

Wei Yang